# Contrasting biosphere responses to hydrometeorological extremes: revisiting the 2010 western Russian Heatwave

Milan Flach[1], Sebastian Sippel[2], Fabian Gans[1], Ana Bastos[3], Alexander Brenning[4,5], Markus Reichstein[1,5], and Miguel D. Mahecha[1,5]

[1]Max Planck Institute for Biogeochemistry, Department of Biogeochemical Integration, P.O. Box 10 01 64, 07701 Jena, Germany
[2]Norwegian Institute of Bioeconomy Research, Ås, Norway
[3]Ludwig-Maximilians University, Department of Geography, Munich, Germany
[4]Friedrich Schiller University Jena, Department of Geography, Jena, Germany
[5]Michael Stifel Center Jena for Data-driven and Simulation Science, Jena, Germany

**Correspondence:** Milan Flach (milan.flach@bgc-jena.mpg.de)

**Abstract.** Combined droughts and heatwaves are among those compound extreme events that induce severe impacts on the terrestrial biosphere and human health. A record breaking hot and dry compound event hit western Russia in summer 2010 (Russian heatwave, RHW). Events of this kind are relevant from a hydrometeorological perspective, but also interesting from a biospheric point of view because of their impacts on ecosystems, e.g., reductions of the terrestrial carbon storage. Integrating both perspectives might facilitate our knowledge about the RHW. We revisit the RHW both from a biospheric and a hydrometeorological perspective. We apply a recently developed multivariate anomaly detection approach to a set of hydrometeorological variables, and then to multiple biospheric variables relevant to describe the RHW. One main finding is that the extreme event identified in the hydrometeorological variables leads to multidirectional responses in biospheric variables, e.g., positive and negative anomalies in gross primary production (GPP). In particular, the region of reduced summer ecosystem production does not match the area identified as extreme in the hydrometeorological variables. The reason is that forest-dominated ecosystems in the higher latitudes respond with unusually high productivity to the RHW. Furthermore, the RHW was preceded by an anomalously warm spring, which leads annually integrated to a partial compensation of 54% (36% in the preceding spring, 18% in summer) of the reduced GPP in southern agriculturally dominated ecosystems. Our results show that an ecosystem-specific and multivariate perspective on extreme events can reveal multiple facets of extreme events by simultaneously integrating several data streams irrespective of impact direction and the variables' domain. Our study exemplifies the need for robust multivariate analytic approaches to detect extreme events in both hydrometeorological conditions and associated biosphere responses to fully characterize the effects of extremes, including possible compensatory effects in space and time.

**Keywords.** compound events, multivariate extreme events, gross primary productivity, heatwaves, droughts, spring-summer compensation.

# 1 Introduction

One consequence of global climate change is that the intensity and frequency of heatwaves will most likely be increasing in the coming decades (Seneviratne et al., 2012). Heatwaves co-occurring with droughts form so-called compound events, for which we can expect severe impacts on the functioning of land ecosystems (e.g., primary production, von Buttlar et al., 2018) that may affect human well-being (e.g., via reduced crop yields, health impacts) (e.g., Scheffran et al., 2012; Reichstein et al., 2013; Lesk et al., 2016). Investigating historical extreme events offers important insights for deriving mitigation strategies in the future.

One well-known example of a compound extreme event is the 2010 western Russian heatwave (RHW). The RHW was one of the most severe heatwaves on record, breaking temperature records of several centuries (Barriopedro et al., 2011). It was accompanied by extensive wild and peat fires with smoke plumes about 1.6 km high at the peak of the heatwave in early August, and estimated emissions of around 77 Tg carbon due to multiple fire events (Guo et al., 2017). Carbon losses due to reduced vegetation activity were estimated to be in the same order of magnitude as losses due to fires (90 Tg, Bastos et al., 2014). The amount of emitted carbon monoxide was almost comparable to the anthropogenic emissions in this region (Konovalov et al., 2011). Approximately 55,000 cases of death have been attributed to health impacts of the RHW (Barriopedro et al., 2011). The RHW was associated with an atmospheric blocking situation (Matsueda, 2011), which lead to a persistent anticyclonic weather pattern in Eastern Europe (Dole et al., 2011; Petoukhov et al., 2013; Schubert et al., 2014; Kornhuber et al., 2016).

However, to fully understand the developments and impacts of heatwaves or droughts, apart from hydrometeorological drivers, associated land-surface dynamics and feedbacks need to be considered (Seneviratne et al., 2010). For instance, under persistent anticyclonic and dry conditions, land-atmosphere feedbacks are expected to further amplify the magnitude of heatwaves via enhanced sensible heat fluxes, as shown also for the RHW (Miralles et al., 2014; Hauser et al., 2016). These feedback mechanisms highlight the importance of depleted soil moisture to heatwaves. In 2010 a negative soil moisture anomaly contributed to increased temperatures (Hauser et al., 2016). It is a general observation that the combination of anticyclonic weather regimes and initially dry conditions prior to the event amplifies heatwaves in most cases (Quesada et al., 2012).

The direct impacts of such extreme events on ecosystems are manifold. Summer heat and drought typically reduce (or even inhibit) photosynthesis, hence reducing the carbon uptake potential of ecosystems (Reichstein et al., 2013). However, the magnitude of these impacts varies between ecosystems (Frank et al., 2015), and the resulting net effects are still under debate, particularly for heatwaves (Sippel et al., 2018). However, in-depth investigations of a number of individual events such as the European heatwave 2003 (Ciais et al., 2005), the 2000-2004 and 2012 droughts in North America (Schwalm et al., 2012; Wolf et al., 2016), and the RHW (Bastos et al., 2014) agree on an overall tendency towards negative impacts on the carbon accumulation potential.

The RHW has been thoroughly investigated from a hydrometeorological point of view linking the atmospheric blocking to the large-scale positive anomalies in air temperatures and negative anomalies in water availability (e.g., Barriopedro et al., 2011; Rahmstorf and Coumou, 2011). The event has been also well investigated with an emphasis on the biospheric impacts describing the negative anomalies in ecosystem productivity and related vegetation indices (e.g., Bastos et al., 2014). However,

comparing the reports of areas affected by the RHW reveals some discrepancies. Hydrometeorological anomalies point at much larger areas affected compared to biosphere response patterns. Fig. 1 shows the zonal evolution of the RHW in both domains. We find that the spatiotemporal patterns of the temperature anomaly does not match the zonal anomaly in vegetation productivity anomalies. Thus, an integrated assessment including the hydrometeorological and the biospheric domain simultaneously may further our understanding of the RHW.

The figure reveals an unusually warm period during spring and one longer heatwave during summertime (Fig. 1a). Temperature anomalies exceed more than 10 K in both spring and summer, but they lead to distinctive anomalies in gross primary productivity (GPP). Positive GPP anomalies occur during the spring event, whereas negative GPP anomalies are occurring during the summer heatwave. The positive GPP response in spring might be a reaction to warmer, more optimal spring temperatures (Wang et al., 2017) possibly accompanied by enough water availability. However, negative GPP anomalies in summer occur only in areas south of 55 $^{\circ}N$ (Fig. 1c) indicating that the GPP response involves much more processes than high temperatures and drought during the unique RHW. As already indicated by Smith (2011), the connection between biosphere and hydrometeorology is much more complex than just a direct one-to-one mapping. Further complicating this issue is the fact that the summer event cannot be investigated without the previous spring as both seasons are inherently related via memory effects in water availability. Increased GPP in spring due to warm temperatures can negatively influence soil moisture and thus GPP during summer (Buermann et al., 2013; Wolf et al., 2016; Sippel et al., 2017). In particular, Buermann et al. (2013) show for North American boreal forests that earlier springs are followed by reduced productivity in summer because of water constraints.

In summary, comparing these two Hovmöller diagrams shows that (1) the affected latitudinal range of the negative GPP anomaly is much smaller than the positive temperature anomaly and (2) the evolution of the summer impacts should consider potential carry over effects of positive GPP anomalies during spring, as earlier studies showed that earlier spring onset and increased spring GPP may negatively influence soil moisture and thus GPP during summer (Buermann et al., 2013). The objective of this paper is to revisit the RHW and to investigate the GPP response during the spring event and the summer heatwave in detail by investigating spatiotemporal anomalies in hydrometeological drivers and ecological variables.

This kind of integrated assessment requires a generic methodological approach. Here, we use a multivariate extreme event detection approach that (1) does not differentiate between a positive and a negative extreme event, and (2) can equally be applied on any set of time series, regardless of whether they describe the biospheric or the hydrometeorological domain. We expect that we can reveal previously overlooked facets in the RHW and discuss whether our approach may facilitate a broader perspective and improved interpretation of extreme events and their impacts.

## 2    Methods & data

### 2.1    Rationale

One approach to detect extreme events like the RHW could be to identify the peaks over some threshold in the marginal distribution of a variable (or its anomaly) of interest. For instance, one could identify values that deviate by more than two

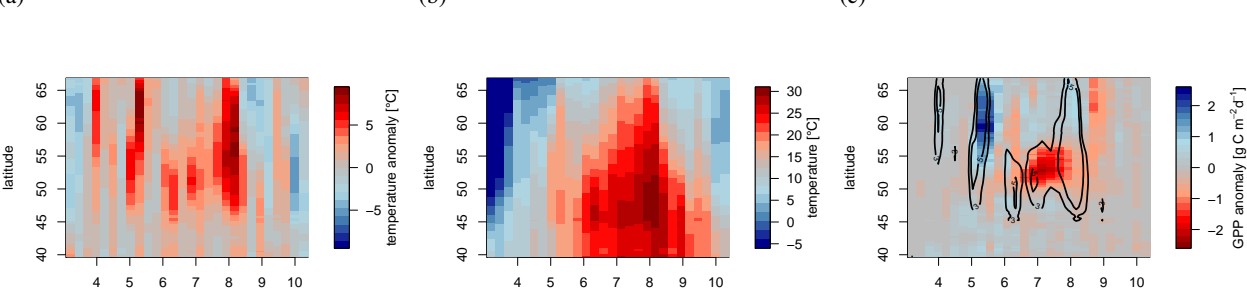

**Figure 1.** Longitudinal average (30.25 to 60.0 ° E) of (a) temperature anomalies (reference period: 2001-2011), (b) absolute temperature, and (c) GPP anomalies in 2010 with a contour of temperature anomalies (+3 K, +5 K).

standard deviations from the variable's mean values (Hansen et al., 2012; Sippel et al., 2015). However, univariate approaches only allow to characterize an event by e.g. extremely high temperature anomalies, lack of precipitation, or very low soil moisture but not their compound anomaly. However, from earlier studies (e.g., Miralles et al., 2014; Hauser et al., 2016) we know that more than one variable is involved in the RHW and a multivariate extreme event detection (i.e., a compound event,

Leonard et al., 2014; Zscheischler and Seneviratne, 2017) is more feasible. Multivariate algorithms to detect extreme events are expected to offer more robust detection capabilities when accounting for dependencies and correlations among the selected variables (e.g., Zimek et al., 2012; Bevacqua et al., 2017; Flach et al., 2017; Mahony and Cannon, 2018). Multivariate extreme event detection considers all observable dimensions of the domain simultaneously. With a multivariate approach one may, for instance, detect very rare combinations of variables even if the individual variables are not extreme. In the following, we detect

the anomalies in a multivariate variable space in two sets of variables describing (1) the hydrometeorological conditions, and (2) the biospheric response. The workflow involves a data pre-processing to compute anomalies, a step for dimensionality reduction to not be biased by redundancies among variables. Based on the reduced data-space, an anomaly score is computed that can then be used as threshold. For various reasons, however, in practice the threshold needs to be computed across multiple spatial grid cells of comparable phenology.

**2.2 Data and pre-processing**

Our dataset for analysing the hydrometeorological domain includes those variables which we consider to be of particular importance for processes taking place during extreme events in the biosphere based on prior process knowledge (Larcher, 2003) and empirical analysis (von Buttlar et al., 2018). The hydrometeorological dataset consists of air temperature, radiation, relative humidity (original resolution $0.71°$, all three from ERA-INTERIM, Dee et al., 2011), precipitation (original resolution

$1°$, Adler et al., 2003), and surface moisture (resolution $0.25°$, http://www.gleam.eu, v3.1a, Miralles et al., 2011; Martens et al., 2017). We consider surface moisture to be a hydrometeorological variable due to its importance for drought detection, although we notice that surface moisture is influenced by biospheric processes. We use gross primary productivity (GPP), latent heat

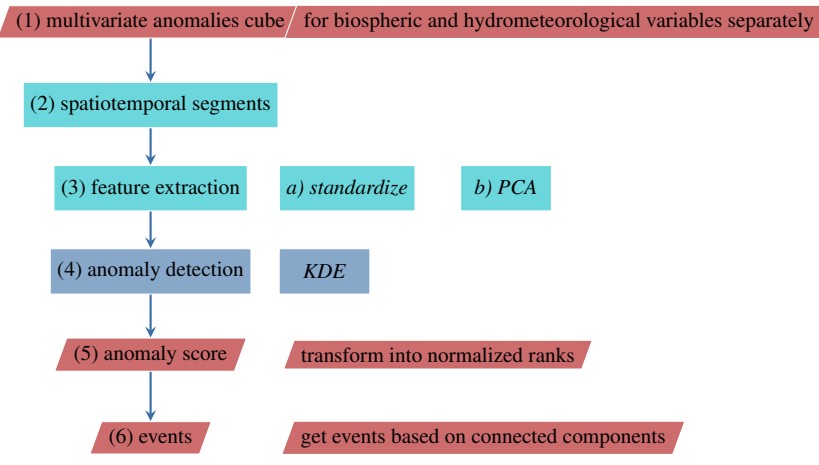

**Figure 2.** Data processing for detecting multivariate anomalies.

flux (LE), sensible heat flux (H) (resolution $0.25°$, all three from FLUXCOM-RS, Tramontana et al., 2016), and the fraction of absorbed photosynthetic active radiation (original resolution $1\ km$, FAPAR, moderate resolution imaging spectroradiometer (MODIS) based FAPAR Myneni et al., 2002) to describe the land surface dynamics.

    The selected variables cover the spatial extent of Europe (latitude $34.5 - 71.5°N$; longitude: $-18 - 60.5°E$) and are re-
gridded on a spatial resolution of $0.25°$ from 2001 to 2011 in an eight-daily temporal resolution. The temporal extend is selected as it is covered by all datasets used in the study. To check for differences in land cover types, we estimate the dominant land cover type of the European Space Agency Climate Change Initiative land cover classification on a spatial resolution $0.25°$ (original: $300\ m$). To check for consistency of our findings among other variables (Sect. 3.2), we additionally use terrestrial ecosystem respiration (TER) and net ecosystem productivity (NEP, both originating from FLUXCOM-RS, Tramontana et al.,
2016).

    The actual event detection is realized on the anomalies of these data sets. To compute the anomalies, For each variable under consideration, we first estimate the seasonality as a smoothed median seasonal cycle per grid cell. We use the median instead of the mean as it is less susceptible to outliers. We then subtract these seasonal cycles from each variable and year to obtain a multivariate data cube of anomalies (Fig. 2, step 1). Small data gaps are set to zeros to ensure that they are not detected
as anomalies. The gap filling is necessary for a multivariate detection approach as there are many more cases in which one variable is missing in the multivariate cube compared to a univariate data stream.

## 2.3   Feature extraction and anomaly detection

We use a multivariate anomaly detection algorithm proposed by Flach et al. (2017) and apply it separately to two sets of variables for the biosphere and hydrometeorology. The method expects a multivariate set of anomalies and projects them to a
reduced space via principal component analysis, retaining a number of principal components that explain more than 95% of

the variance (Fig. 2, step 3b). This procedure accounts for linear correlations in the data only by removing redundancies among the variable anomalies.

We compute an anomaly score via kernel density estimation (KDE, Parzen, 1962; Harmeling et al., 2006) in the reduced anomaly space (Fig. 2, step 4). KDE showed very good performance among different other options to detect multivariate anomalies in previous experiments (Flach et al., 2017). One strength of KDE is that it considers nonlinear dependencies among dimensions (Fig. 3). The anomaly scores are transformed into normalized ranks between 1.0 (very anomalous, data point in the margins of the multivariate distribution) and 0.0 (completely normal, data point in the dense region of the multivariate distribution; Fig. 2, step 5). In this univariate index of compound extremes, it is legitimate to use a classical threshold that can be intuitively analysed. However, to avoid an equal spatial distribution of event occurrences we do not apply this multivariate anomaly detection per pixel, but rather by region.

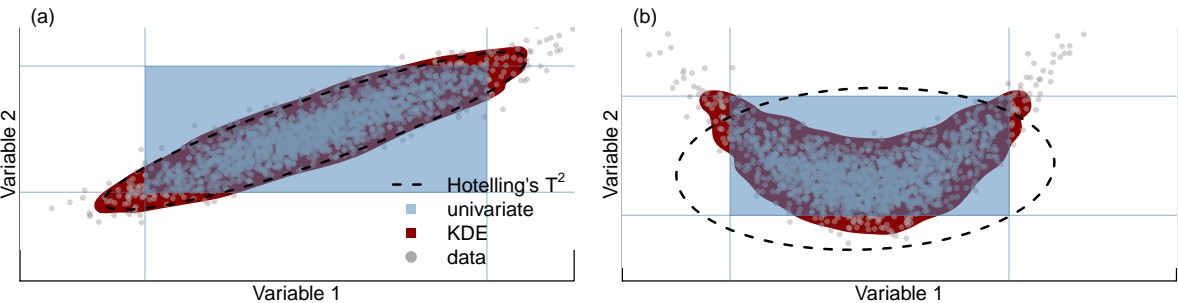

**Figure 3.** Illustration of the multivariate anomaly detection algorithm with two variables. The data has: (a) linear dependencies (multivariate normal) and (b) a nonlinear dependency structure. Univariate extreme event detection (peak-over-threshold in the marginal distribution of a variable) does not follow the shape of the data, whereas algorithms assuming a multivariate normal distribution (Hotelling's $T^2$, Lowry and Woodall, 1992) are suitable for case (a); kernel density estimation (KDE) gets the shape of the data in both cases (a) and (b). 5% extreme anomalies are outside the shaded areas (region of "normality") for all three algorithms.

## 2.4 Spatiotemporal segmentation

The spatiotemporal segmentation aims to identify spatial areas of comparable phenology, climate and seasonality. For identifying these regions, we follow the methodology described by Mahecha et al. (2017) and extend it to the multivariate case. The main idea is that the (now spatial) principal components of the mean seasonal cycles can be used for classifying regions according to their characteristic temporal dynamics.

The procedure for extracting spatial segments of similar grid cells works as follows (for a detailed description see Mahecha et al., 2017):

(1) We estimate the median seasonal cycle in each grid cell and of each variable individually and standardize the median seasonal cycles to zero mean and unit variance to get the cycles comparable across different units (Fig. 4 (1)).

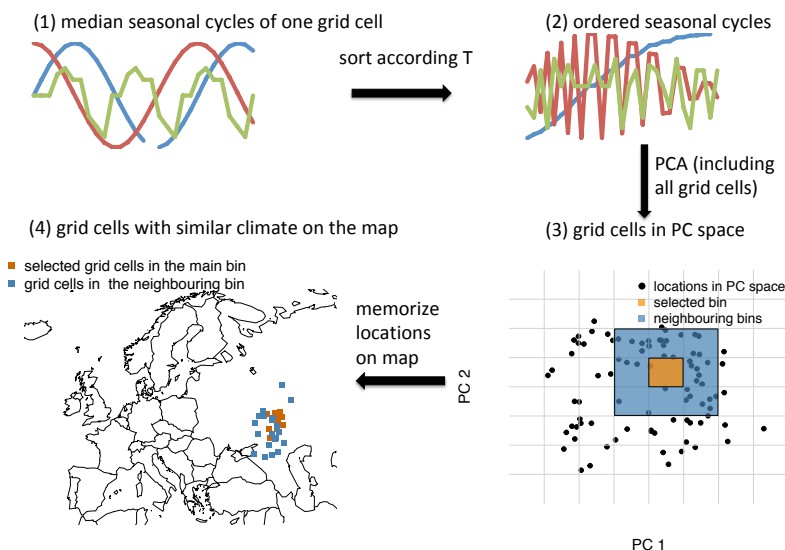

**Figure 4.** Illustration of the spatial segmentation procedure with two principal components.

(2) To remove the effect of different phasing (similar, but only lagged seasonal cycles), we sort the median seasonal cycles according to a variable showing a strong seasonality, which is temperature in our case. Thus, we memorize how to bring temperature in a sorted increasing or decreasing order (the 'permutation' of temperature) and apply the same permutation to the other median seasonal cycles (Fig. 4 (2)). We prepare the data for dimensionality reduction by concatenating the seasonal cycle of all variables to a matrix seasonal cycles × space. We apply a principal component analysis (PCA) to reduce the dimension of the concatenated median seasonal cycles.

(3) We select locations (grid cells) of similar phenology and climate by dividing the orthogonal principal component subspace into equally sized bins (Fig. 4 (3)). We used $N_{PC} = 4$ components in this step, explaining 71% of variance. The bins are sufficiently small compared to the length of the principal components to ensure a fine binning of very similar phenology and climate.

(4) We compute the multivariate anomaly score in an overlapping moving window for all grid cells that fall into one of the bins (the central bin and the neighbouring bins, Fig. 4 (4)).

A final detail to consider is the effect of changing seasonal variance (temporal heteroscedasticity). These patterns lead to detecting extreme events predominantly during the high-variance seasons (i.e. summer times). To avoid seasonal biases in the extreme event detection, we additionally apply the entire anomaly detection scheme to seasonally overlapping moving windows across years.

Within the spatiotemporal segmentation procedure, we ensure that the number of observations is at least 198 (9 time steps × 11 years, at least one spatial replicate). To reunify the spatiotemporal segments, we assign the normalized anomaly scores

temporally to the time step in the center of the temporal moving window and spatially to the grid cell in the central bin of similar climate and phenology.

## 2.5 Statistics of extreme events

We assume that 5% of the data are anomalous in each overlapping spatiotemporal segment and convert the anomaly scores into
binary information. However, the main results of compensation effects are not sensitive to this threshold selection (Appendix
Tab. A1, varying the threshold between 1% to 10%). To compute statistics based on the spatiotemporal structure of each extreme
event, we follow an approach developed by Lloyd-Hughes (2011); Zscheischler et al. (2013) and compute the connections
between spatiotemporal extremes if they are connected within a $3 \times 3 \times 3$ (lon $\times$ lat $\times$ time) cube. Each connected anomaly
is considered as a single event (Fig. 2, step 6). In this way, we observe event-based statistics, i.e., affected area (km$^2$), affected
volume (km$^2 \cdot$ days), centroids of the area and histograms of the single variable anomalies stratified according to different
ecosystem types (land cover classes). Furthermore, we observe the response of individual variables to the multivariate event
by computing the area weighted sum of the variable during the event in which the variable of interest is positive relative to the
seasonal cycle ($res^+$) or negative, respectively ($res^-$). For many biospheric variables, one expects a mainly negative response
to hydrometeorological extreme events like heatwaves or droughts (Larcher, 2003; von Buttlar et al., 2018). Thus, we define
compensation of a specific variable to be the absolute fraction of $res^+$ from $res^-$. The balance of a variable is the sum of $res^+$
and $res^-$. Centroids of $res^+$ and $res^-$ are computed as average of the affected longitudes, latitudes, and time period, weighted
with the number of affected grid cells at this longitude, latitudes, and time period, and its respective anomaly score. They are
used to compute the spatial and temporal distance between $res^+$ and $res^-$. Affected area, volume, response and centroids take
the spherical geometry of the Earth into account by weighting the affected grid cells with the cosine of the respective latitude.

## 3 Results

### 3.1 Extreme events in western Russia in 2010

We identify two multivariate extreme events in the set of hydrometeorological variables in western Russia 2010, based on the
spatiotemporal connectivity. The two extreme events are separated by approximately one week of normal conditions towards
the end of May:

- hydrometeorological spring event: anomaly of the hydrometeorological variables in western Russia during May ranging
  from longitude 30.25 - 60.0 ° E, latitude $\geq 55°N$ (Fig. 5a, b)
- hydrometeorological summer event: anomaly of the hydrometeorological variables in western Russia, June to August,
  ranging from longitude 28.75 - 60.25° E, latitude 48.25 - 66.75 °$N$. This event is usually referred to as Russian Heatwave
  (RHW) 2010 (e.g., Barriopedro et al., 2011; Rahmstorf and Coumou, 2011) (Fig. 5c, d).

Both multivariate hydrometeorological anomalies partly overlap with a multivariate anomaly in the set of biosphere variables
(biospheric spring event and biospheric summer event). Of specific interest is that the area affected by anomalous hydrometeo-

rological summer conditions is remarkably larger than the one detectable in the biospheric variables (biospheric summer event, $2.4 \cdot 10^6$ vs. $1.1 \cdot 10^6$ km$^2$, Tab. 1). This fact already indicates that biosphere responses are more nuanced than the hydrometeorological events and do not simply follow the extent of the hydrometeorological anomaly. As indicated e.g., also by Smith (2011), a hydrometeorological extreme event does not necessarily imply an extreme response.

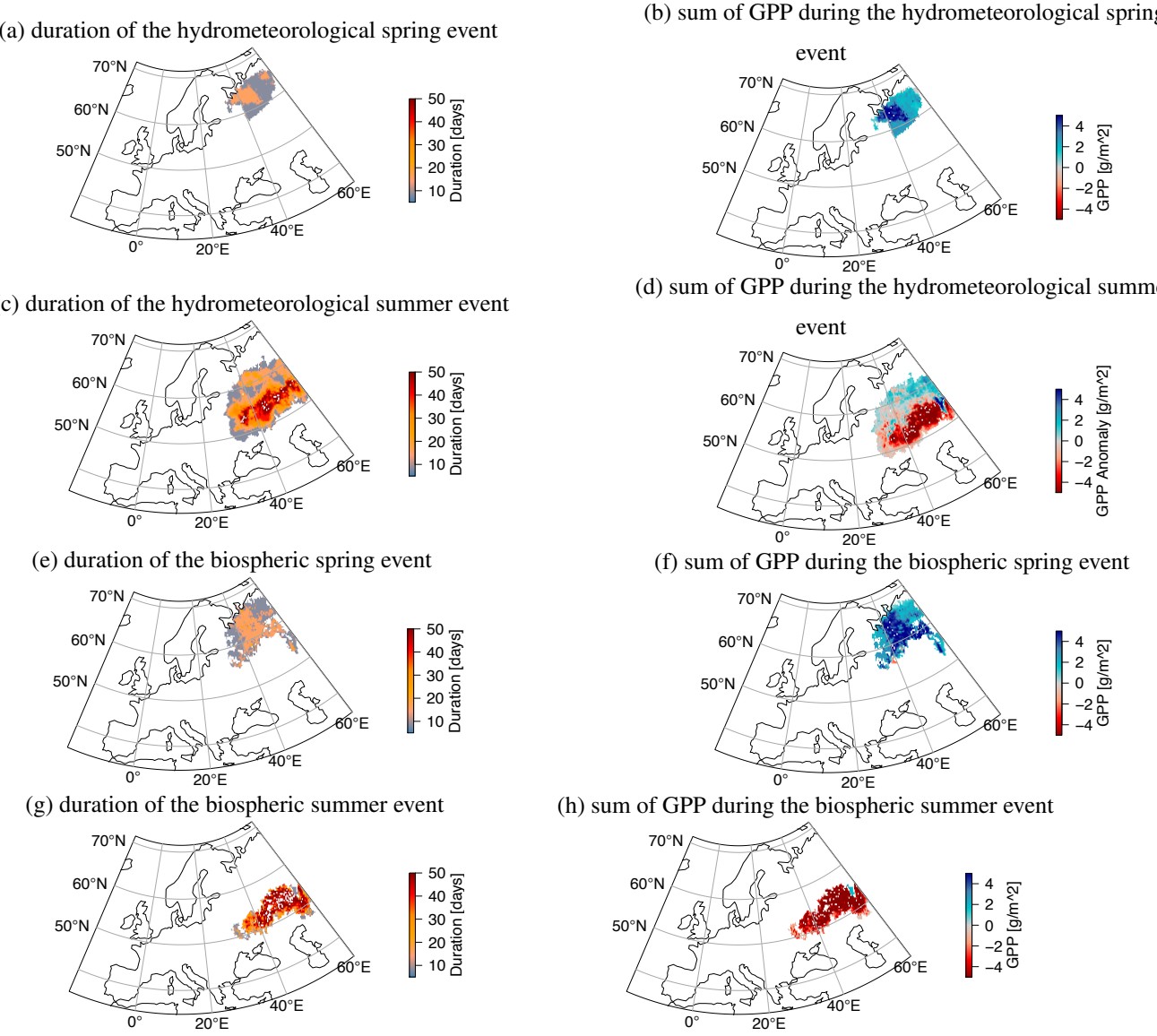

**Figure 5.** Left column: temporal duration of the (a) hydrometeorological spring event, (c) hydrometeorological summer event, and biospheric events (e),(g). Right column: corresponding GPP response, i.e., the sum of deviations from the seasonal cycle during the event for the (b) hydrometeorological spring event, (d) hydrometeorological summer event, and biospheric events (f), (h). While the GPP response during the hydrometeorological spring event is entirely positive (more productive than usual, b), GPP response during the hydrometeorological summer event differs between higher latitudes ($> 55°$ N, short-lasting, positive) and lower latitudes (long-lasting, negative).

### 3.1.1 Hydrometeorological events

As GPP is a key determinant of ecosystem–atmosphere carbon fluxes, we focus on the gross primary productivity (GPP) response to the multivariate hydrometeorological anomaly: We find that the GPP response is entirely positive during the short-lasting hydrometeorological spring event (+17.8 Tg C, Tab. 1), while it is mainly negative during the summer event (+8.8 Tg C, −49 Tg C, Tab. 1). A part of the GPP summer losses (18%) associated with the RHW in the southern region are instantaneously reduced by over-productive vegetation in the higher latitudes, which are hit by the extreme event. Please note, that the carbon balance in summer accounts for the GPP response to the same hydrometeorological extreme event, namely the RHW, which leads to contrasting responses in adjacent regions. If we estimate the annually integrated effect of the anomalies, another 36% of the carbon losses are compensated during spring in higher latitudes.We did not find extreme events after summer, which implies a fast recovery of vegetation activity after summer. Integration over the spring and summer events thus equals the annual integration. Overall, we find that 54% of the negative GPP anomalies are compensated either because of the positive spring anomalies or across ecosystems hit by the same event during summer. These compensation effects reduce the negative carbon impact of integrated annual hydrometeorological event from −49.0 Tg C to −24 Tg C in total (Tab. 1). We want to emphasize, that the negative impact of the RHW in therms of GPP is just reduced, and still negative in total.

### 3.1.2 Biospheric events

Moving the focus to the multivariate biosphere events (biospheric spring and biospheric summer event), which overlap with the hydrometeorological events, we find that GPP responses based on the biospheric spring event are almost entirely positive (+33.8 Tg C), and based on the biospheric summer event almost entirely negative (−82.6 Tg C). If we consider the annually-integrated effect of the anomalies, spring carbon gains are estimated to offset 41% of the subsequent carbon losses in summer (56 days earlier) in the higher latitudes (514 km distance of the centroids, Tab. 1). To further examine these findings, we check for these kind of compensation effects among different variables and another GPP dataset in the following section. Note that the dataset of biosphere variables includes GPP itself. Computing the responses based on the extent of the biospheric event is nevertheless useful, as an extreme event in the biosphere variables is not exclusively restricted to extreme conditions in the hydrometeorological conditions (Smith, 2011).

### 3.2 Compensation in other data-sets and variables

The annually-integrated compensation effect in GPP is highly consistent among different variables. For instance, NEP (excluding fire) shows such kind of compensation, but also FAPAR and LE (Tab. 2). Sensible heat flux, on the other hand, is high during the hydrometeorological summer event (biospheric summer event), as well as the hydrometeorological spring event (biospheric spring event) as expected for strong positive temperature anomalies. However, some of the remote sensing data products might be affected by high fire induced aerosol loadings during the heatwave that affect atmospheric optical thickness (e.g., Guo et al., 2017; Konovalov et al., 2011). Exploring an almost entirely climate-driven GPP product (FLUXCOM

**Table 1.** Statistics of the extreme events, based on their spatiotemporal connectivity structure: affected area, affected volume, positive and negative GPP response ($res^{+/-}$) to the event, compensation of the negative response (comp.), as well as average spatial and temporal distance between the parts of the events with positive and negative responses.

| event | area $[km^2]$ | volume $[km^2 \cdot days]$ | GPP comp. | $res^+_{GPP}$ | $res^-_{GPP}$ | spatial [km] | temporal [d] |
|---|---|---|---|---|---|---|---|
| **hydrometeorological** | | | | | | | |
| spring | $0.77 \cdot 10^6$ | $0.81 \cdot 10^7$ | - | $17.8\,Tg$ | - | | |
| summer | $2.44 \cdot 10^6$ | $5.79 \cdot 10^7$ | 0.18 | $8.8\,Tg$ | $-49.0\,Tg$ | 499 | -4 |
| integrated | $3.29 \cdot 10^6$ | $6.60 \cdot 10^7$ | 0.56 | $26.6\,Tg$ | $-49.0\,Tg$ | 452 | -34 |
| **biospheric** | | | | | | | |
| spring | $1.25 \cdot 10^6$ | $1.48 \cdot 10^7$ | 117.04 | $33.8\,Tg$ | $-0.3\,Tg$ | 756 | -16 |
| summer | $1.06 \cdot 10^6$ | $4.22 \cdot 10^7$ | 0.00 | $0.4\,Tg$ | $-82.4\,Tg$ | 962 | 50 |
| integrated | $2.28 \cdot 10^6$ | $5.70 \cdot 10^7$ | 0.41 | $34.2\,Tg$ | $-82.7\,Tg$ | 514 | -56 |

**Table 2.** Negative responses to the RHW are partly compensated based on the integrated biospheric or hydrometeorological events in 2010. The finding is consistent over different variables and data sets.

| | hydrometeorological events | | | biospheric events | | |
|---|---|---|---|---|---|---|
| variable | $res^+$ | $res^-$ | comp. [%] | $res^+$ | $res^-$ | comp. [%] |
| NEP | $17.53\,Tg$ | $-34.03\,Tg$ | 51.5 | $23.45\,Tg$ | $-48.49\,Tg$ | 48.4 |
| LE | $19.90\,Tg$ | $-53.97\,Tg$ | 36.9 | $16.34\,Tg$ | $-102.81\,Tg$ | 15.9 |
| FAPAR | 1.89 | $-4.03$ | 47.0 | 2.52 | $-6.61$ | 38.1 |
| TER | $18.97\,Tg$ | $-11.06\,Tg$ | 171.4 | $13.71\,Tg$ | $-23.43\,Tg$ | 58.5 |

RS+METEO, Jung et al., 2017), we also find the integrated compensation effect, although much lesser pronounced (Appendix Fig. B1). Thus, we are confident that the observed compensation effect is not related to the optical thickness during the RHW.

### 3.3 Influence of Vegetation Types

In Fig. 6 we present the histograms of GPP anomalies for different land cover classes (forests, grasslands and crops) based on the hydrometeorological spring event, and hydrometeorological summer event (biospheric spring event, and biospheric summer event, respectively, Fig. C1) to highlight two aspects: First, during the spring event (hydrometeorological spring or biospheric spring), forests react almost entirely with positive GPP anomalies (Fig. 6a). Forests in this region are energy-limited, so the timing of the extreme event leads to hydrometeorological conditions (e.g., positive temperature anomalies in spring, more incoming radiation accompanied by enough water availability) which are favourable for vegetation productivity, as absolute spring temperatures are still below the temperature optimum of GPP (Fig. 8a, Wolf et al., 2016; Wang et al., 2017).

Second, during the hydrometeorological summer event, we observe positive to neutral GPP responses in forests, whereas crops and grasslands react strongly negative (Fig. 6b). The positive versus negative GPP responses almost entirely reflect the

(a) hydrometeorological spring event          (b) hydrometeorological summer event

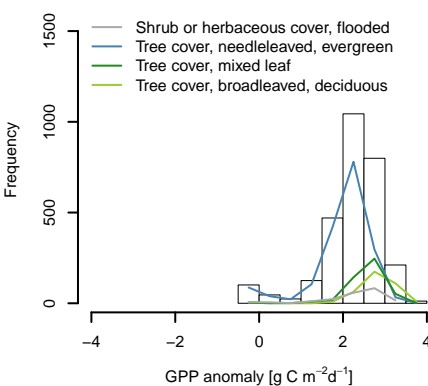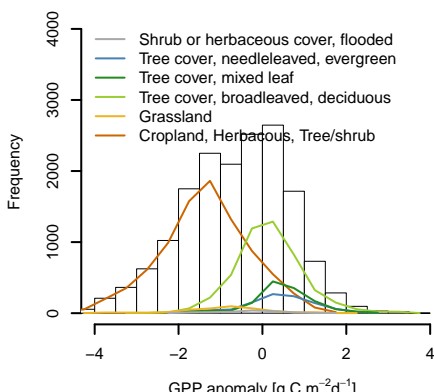

**Figure 6.** Histogram of GPP anomalies (reference period: 2001-2011) for different land cover classes based on the spatio–temporal extent of (a) the hydrometeorological spring event and (b) the hydrometeorological summer event. Bars denote the sum of all vegetation classes.

(b) sum of GPP during the hydrometeorological summer event

(a) Land Cover

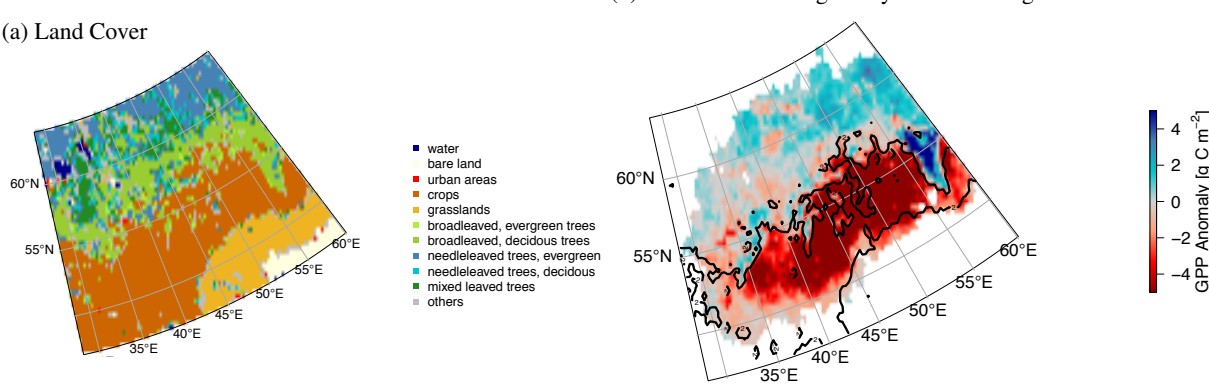

**Figure 7.** (a) Dominant land cover classes of a spatial extent of the RHW. (b) The boundaries of the different ecosystem types (forest-dominated ecosystems vs. agriculture-dominated ecosystems, denoted by the black contour line) match the observed patterns of the GPP response (reference period for the calculating anomalies: 2001-2011) during the hydrometeorological summer event.

map of dominant vegetation types (forest vs. agricultural ecosystems, Fig. 7). However, different vegetation types exhibit a transition from higher latitudes (predominantly forest ecosystems) to lower latitudes (dominated by agricultural ecosystems). Thus, the different responses of vegetation types might be confounded by the fact that absolute temperatures also follow a latitudinal gradient (Fig. 1b). Absolute temperatures for agricultural ecosystems are higher and far beyond the temperature optimum of GPP (Fig. 8c). Additionally, agricultural ecosystems are drying out in summer (low soil moisture, Fig. 8c). In contrast, forest-dominated ecosystems at higher latitudes experience temperatures just slightly above the temperature optimum of

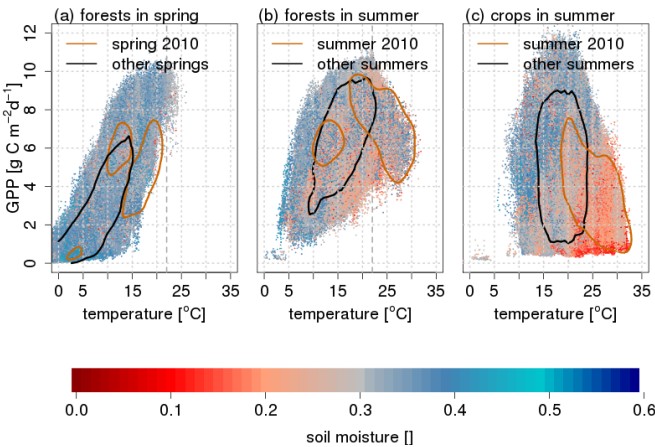

**Figure 8.** Temperature optimality for GPP in (a) forests during spring, (b) forests during summer, and (c) crops during summer. Contour lines enclose 75% of the data points.

GPP, accompanied by high soil moisture (Fig. 8b). The response of forest ecosystems partly reflects a latitudinal gradient: forest ecosystems in the lower latitudes react positively to the spring temperature anomaly and then tend to react more negatively to the summer heatwave than forest ecosystems in higher latitudes. Forest ecosystems in higher latitudes are still productive in terms of GPP during the peak of the heatwave (Fig. 9). We find negligible anomalies in autumn for both ecosystems, which implies a fast recovery after the heatwave.

To disentangle the variable importance of the different confounding factors, we run a simple linear regression model which tries to explain GPP as function of the hydrometeorological driver variables (temperature, precipitation, radiation, surface moisture, including their anomalies and absolute values), as well as vegetation type, duration and latitude (Appendix D). We use an algorithm after Chevan and Sutherland (1991) which extracts the independent contribution of the variable importance related to this particular variable regardless of the model complexity or dependencies among variables. The model reveals from a statistical point of view, that vegetation type and the latitudinal gradient are the most important variables explaining GPP during the summer event, followed by the hydrometeorological drivers. Access to deeper water and soil type as well as non-linear feedbacks are factors which are not represented in the model, but might explain the high importance of latitude. Apart from vegetation type being important for the GPP response, underlying water use efficiency (calculated according to Zhou et al. (2014) is consistently higher in forest-dominated ecosystems compared to agriculture-dominated ecosystems (Appendix Fig. E1a), and higher evaporative fraction in forest ecosystems during the peak of the heatwave (Appendix Fig. E1b).

## 4   Discussion

In this paper we show that the hydrometeorological extreme events affecting western Russia in spring and summer 2010 are not directly mapping to the observed vegetation responses. Positive to neutral GPP responses prevail in higher latitudes during

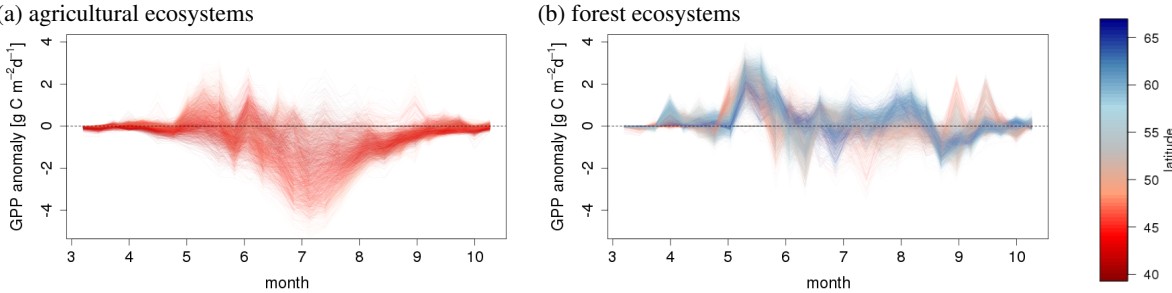

**Figure 9.** Temporal evolution of the GPP anomaly (reference period: 2001-2011) for (a) agricultural ecosystems and (b) forest ecosystems, colored according to the latitude.

summer, whereas strong negative impacts on GPP can be found in lower latitudes. We interpret this effect by different water management strategies of forest vs. agricultural ecosystems (Teuling et al., 2010; van Heerwaarden and Teuling, 2014) that meet a general latitudinal temperature gradient. Apart from a more efficient water usage of forest-dominated ecosystems, access to deeper soil water might be another reason of ecosystem-specific responses (Fan et al., 2017; Yang et al., 2016). Note that the latitudinal temperature gradient alone might explain differences in the response within ecosystems in summer and between spring and summer, but does not sufficiently explain differentiated GPP responses in summer among different ecosystems (predominantly forest vs. agricultural ecosystems).

Another important aspect is that the combination of the anomalous spring and the unique heatwave in summer might be inherently connected via land surface feedbacks. Buermann et al. (2013) showed that warmer springs going in hand with earlier vegetation activity negatively affect soil moisture in summer, and thereby vegetation activity. It is a general observation that warm and dry springs enhance summer temperatures during droughts, which suggests the presence of soil-moisture temperature feedbacks across seasons (Haslinger and Blöschl, 2017). In case of the Russian heatwave 2010, soil moisture was one of the main drivers (Hauser et al., 2016), in hand with persistent atmospheric pressure patterns (Miralles et al., 2014). Thus, we suspect that the spring event is connected to the summer heatwave in 2010, if not setting the preconditions for a heatwave of this unique magnitude.

The integration of the carbon balance over spring and summer might be justified by assumed connections between spring and summer as outlined before. However, we would like to note that a common annual integration and assessment of compensatory effects on the carbon balance over events during the growing season equals the integration over spring and summer for this particular case, as we did not find any events after summertime. The absence of events after the summer heatwave implies a fast recovery of the ecosystems.

Compensations of parts of the negative impacts on the carbon balance during hydrometeorological extreme events have been reported in earlier studies. On the one hand, Wolf et al. (2016) report that a warm spring season preceding the 2012 US summer drought reduced the impact on the carbon cycle. Yet on the other hand, the increased spring productivity amplified the reduction in summer productivity by spring–summer carry-over effects via soil moisture depletion: higher spring productivity

leads to higher water consumption in spring. The high water additionally consumed during spring reduces the water availability in summer and thereby affects productivity during the following summer. However, it remains unclear whether this observation was a singular case, or whether it could become a characteristic pattern to be regularly expected in a warmer world. In this study, we provide some evidence for presumed comparable effects. In contrast to the discussion in Wolf et al. (2016), enhanced

productivity does not exclusively occur temporally, i.e., spring partly compensates for summer losses, but rather spatially adjacent forest ecosystems are reducing the negative impact of agricultural ecosystems on the carbon balance. Spatially adjacent ecosystems partly compensating carbon losses due to drought or heatwaves have been observed earlier, e.g., in mountainous ecosystems that respond differently than lowlands during the European heatwave 2003 (Reichstein et al., 2007).

    Following up on compensatory effects, Sippel et al. (2017) use ensemble model simulations to disentangle the contribution of

spring compensation vs. spring–summer carry-over effects on a larger scale. They show that in general warm springs compensate for parts of summer productivity losses in Europe, whereas spring–summer carry-over effects are constantly counteracting by enhancing summer losses. Also Mankin et al. (2017, 2018) note that increased spring productivity with spring–summer carry-over effects can be observed in earth system models. We can confirm the general finding that spring partly compensates for summer productivity losses in observations for our case study on the RHW. Without using model simulations it is difficult

to quantify spring–summer carry-over effects via soil moisture depletion. In case of the RHW only very few areas are anomalously productive in terms of GPP in spring and unproductive in summer as well. Thus, we suspect that exclusively temporal spring–summer carry-over effects play a rather small role for the RHW. However, we also emphasize that longer-term effects, such as effects in subsequent years through species changes for instance (Wagg et al., 2017), have not been considered in the present study and likely remain hard to quantify beyond dedicated experiments.

The RHW is among the best studied extreme events in the Northern Hemisphere. However, the enhanced productivity of Northern forests which diminishes the negative carbon impact of the RHW as reported in this study has only received marginal attention so far. For instance, Wright et al. (2014) mention positive NDVI anomalies in spring 2010, but then focus largely on productivity losses in the Eurasian wheat belt. Similarly, Bastos et al. (2014) focus on a spatial extent of the biosphere impacts that only partly includes forest ecosystems at higher latitudes. Our estimation of carbon losses due to decreased vegetation

activity (82 Tg C) is comparable to the one of Bastos et al. (2014) (90 Tg C). Similar to the results of our study, Yoshida et al. (2015) report reductions in photosynthetic activity in agriculture-dominated ecosystems during the RHW, but only small to no reductions in forest ecosystems during summertime. However, their interpretations focus on the summer heatwave. Nevertheless, re-evaluating impact maps (published e.g., in Wright et al., 2014; Yoshida et al., 2015; Zscheischler et al., 2015) in the light of our findings suggests that their evidence supports the presence of contrasting responses, differing among

ecosystems during the RHW. When it comes to extreme events, the general tendency in many existing studies is naturally to focus on negative impacts as they are of particular interest for society (Bastos et al., 2014; Wright et al., 2014; Yoshida et al., 2015; Zscheischler et al., 2015).

## 5 Conclusions

We re-analysed biospheric and hydrometeorological conditions in western Russia 2010 with a generic spatiotemporal multivariate anomaly detection algorithm. We find that the hydrometeorological conditions and the biospheric responses exhibit two anomalous extreme events, one in late spring (May) and one over the entire summer (June, July, August), covering large areas of western Russia. For the summer event, we find that the spatially homogeneous anomaly pattern (characterized by high solar radiation and temperature, low relative humidity and precipitation) translate into a bimodal and contrasting biosphere response. Forest ecosystems in higher latitudes show a positive anomaly in gross primary productivity, while agricultural systems decrease their productivity dramatically.

If we consider the annually integrated effect of the anomalous hydrometeorological conditions in 2010, we find that forest ecosystems reduce the negative impact of the productivity losses experienced in agricultural ecosystems by 54% (36% during spring, 18% during summer). Please note, that the annually integrated impact of the 2010 events on the carbon balance stays strongly negative. Our findings do not alleviate the consequences of extreme events for food security in agricultural ecosystems.

From a methodological point of view, this study emphasizes the importance of considering the multivariate nature of anomalies. From this study, we learn that it is insightful to consider both, the possibility of negative as well as of positive impacts, and assess their annually integrated statistics. Although the integrated impact of gross primary production on the hydrometeorological conditions in 2010 is strongly negative, it is important to notice the partial compensatory effects due to differently affected ecosystem types, as well as timing of the extreme events.

## Appendix A: Sensitivity of the threshold selection

**Table A1.** Compensation effects of the integrated hydrometeorological events (spring and summer) are not sensitive to varying the threshold for extreme event detection between 93% to 99% (7% to 1% of extreme data in each spatiotemporal segment). A slight tendency towards more pronounced compensation effects can be seen for the 90% threshold. Such kind of enhancing the positive response is expected for lower thresholds, as the hydrometeorological conditions are not perceived as "extreme" anymore.

| | Compensation [%] | | | | |
|---|---|---|---|---|---|
| Threshold | 90% | 93% | 95% | 97% | 99% |
| GPP | 65 | 53 | 54 | 58 | 55 |
| NEP | 60 | 52 | 52 | 51 | 46 |
| LE | 49 | 36 | 37 | 38 | 32 |
| FAPAR | 70 | 46 | 47 | 50 | 50 |
| TER | 150 | 147 | 171 | 191 | 197 |

## Appendix B: Comparison with METEO + RS

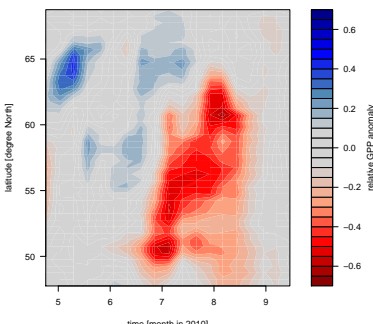

**Figure B1.** The longitudinal (30.25-60.25° E) average of the GPP anomalies during the RHW 2010, based on the Climate Research Unit observation-based climate variables (CRUNCEPv6, New et al., 2000) driven GPP product originating from FLUXCOM RS+METEO (Jung et al., 2017) shows similar but weaker compensation effects. 28% of the negative GPP response to the RHW are compensated based on the shown latitude-longitude subset.

## Appendix C:  Biosphere response

(a) biospheric spring event

(b) biospheric summer event

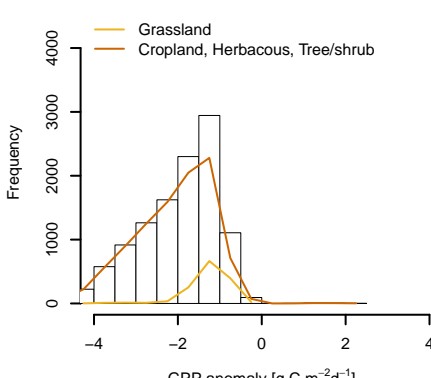

**Figure C1.** Histogram of GPP anomalies (reference period: 2001-2011) for different land cover classes constrained by a) biospheric spring and b) biospheric summer event.

## Appendix D:  Factors explaining the GPP response

As several factors might contribute to the GPP response to the hydrometeorological anomalies in spring and summer 2010, we assume that a linear model can partly explain the variance in GPP and improve our understanding of the extreme events in
spring and summer via the variable importance of the model. Thus, we model GPP of all pixels during spring and separately during summer as a function of the factors temperature (T), precipitation (P), global radiation (Rg), soil moisture (SM), and their corresponding anomalies. We include land cover type, duration and latitude as possible drivers of the full model (spring $R^2$ = 0.86, summer $R^2$ = 0.35). We use a variable importance partitioning algorithm according to Chevan and Sutherland (1991) to get the variable importance of the full model while accounting for redundancies (e.g., dependencies) among the
factors and model complexity. The partitioning algorithms computes all possible combinations of submodels (excluding one or several factors). By combining the differences of $R^2$ measures of the submodels in an intelligent way (for more details see Chevan and Sutherland, 1991), it is possible to partition the total importance of each variable into an independent contribution and a joint contribution. Results show, that the hydrometeorological spring event is mainly a response to very favourable hydrometeorological conditions (higher radiation due to the lack of precipitation, high absolute spring temperatures beyond
the optimum of GPP), which is indicated by the high independent contributions of the variables. As only forest ecosystems are affected, vegetation type plays a minor role (Fig. D1 a). The lower explanatory power of the model for the summer event indicates that there are potentially non-linear feedback loops not captured by the model or factors playing a role, which we did not include in the model. One of the latter candidates is the access to deeper water, also indicated by the high variable

importance of latitude. Apart from latitude vegetation type is the most important factor driving the GPP respose during the summer event (Fig. D1 b).

(a)

(b)

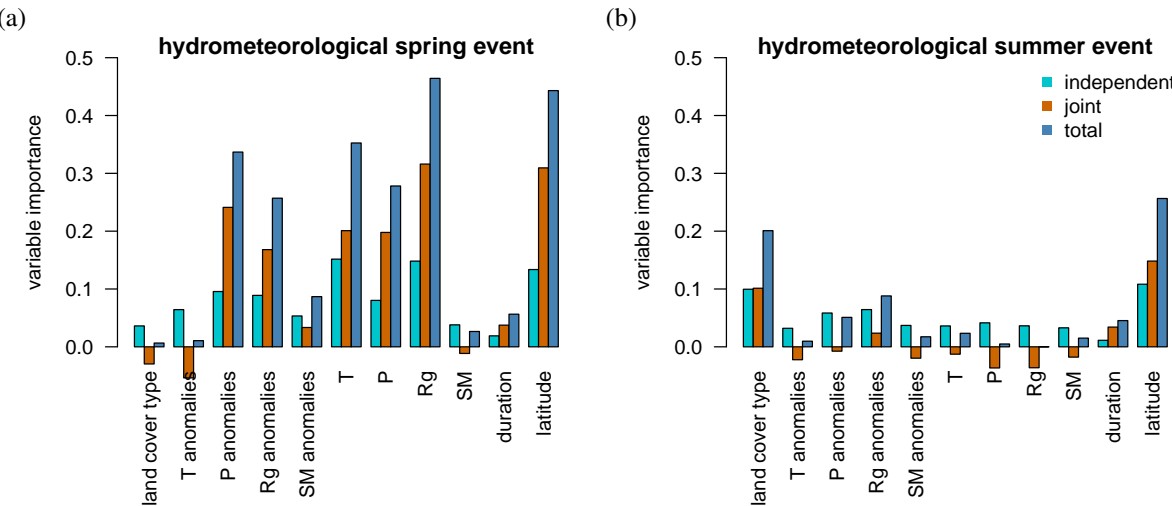

**Figure D1.** Independent, joint and total contribution of the factors explaining (a) GPP response during the hydrometeorological spring event and (b) during the hydrometeorological summer event. Used abbreviations are: T (temperature), P (precipitation), Rg (radiation), SM (soil moisture).

## Appendix E: Water use efficiency and evaporative fraction of different land cover types

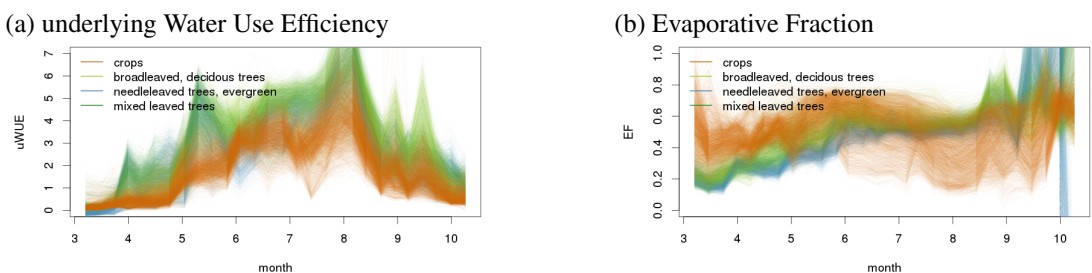

**Figure E1.** (a) Underlying water use efficiency (uWUE) and (b) evaporative fraction (EF) of the area affected by the RHW in 2010. uWUE is calculated according to Zhou et al. (2014) including vapour pressure deficit. In contrast to WUE, uWUE attempts to correct for differences in temperature and vapour pressure deficit to a certain degree.

*Author contributions.* MF and MDM designed the study in collaboration with SS, FG, ABa, ABr and MR. MF conducted the analysis and wrote the manuscript with contributions from all co-authors.

*Competing interests.* The authors declare that they have no conflict of interest.

*Acknowledgements.* This research received funding by the European Space Agency (project "Earth System Data Lab") and the European Union's Horizon 2020 research and innovation programme (project "BACI", grant agreement no 64176). The authors are grateful to the FLUXCOM initiative (http://www.fluxcom.org) for providing the data. MF acknowledges support by the International Max Planck Research School for Global Biogeochemical Cycles (IMPRS). Furthermore, the authors would like to thank Sebastian Bathiany for crucial discussions on the topic, Jürgen Knauer for his expertise on water use efficiency, Julia Kiefer for her kind language check, as well as Victor Brovkin and Sophia Walther helping to improve the manuscript. Two anonymous reviewers provided valuable suggestions for improvement.

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
