# Peer review of "Contrasting biosphere responses to hydrometeorological extremes: revisiting the 2010 western Russian Heatwave"

_Biogeosciences, 2018_

## Referee Comment (RC1) · Anonymous Referee #1 · 24 Apr 2018

Summary:

This manuscript presents a case study analysis to examine the impacts of compound events through a comparison of hydrological (via soil moisture) and biospheric (via GPP) perspectives in the season preceding, and during, the Russian 2010 heatwave. The paper provides a case for why singular extreme events need to be examined under different perspectives to understand the full implications of these events across multiple sectors. It is a nice study however I was anticipating a more indepth analysis of the processes that connect the two events. Its almost there and perhaps only requires minor revision of the text to achieve this.

[Figure]

Main Comments:

1) The hydrological event and the biospheric events don't have the same spatial coverage which makes it hard for those new to the concept of compound events to appreciate how the events evaluated in the manuscript are indeed related. Could the authors perhaps provide a stronger case for why these distinctive events should be considered together beyond the 'different disciplinary perspectives' by delving into how one may be a result of the other. The commentary around Figure 1 on page 3 makes it difficult to reconcile the fact that the two events are related. Perhaps part of the confusion also stems from having a spring event, a summer event and then considering these events defined in terms of either the biospheric and hydrological perspective (so effectively giving 4 events to compare). I think this can be resolved by amending the text and including more discussion on how these events fit together.

2) The narrative in section 2.2 was hard to follow in that there is some information that may be better to remove (e.g. defining extremes using global thresholds) or a dependence on jargon that not everyone may understand (some examples noted in the minor comments). Given that the manuscript aims to articulate a methodology for extracting information on compound events this could be revised. Would it be possible to add some illustration to the schematic in Figure 2 to clarify how the spatiotemporal segments are defined and extracted.

3) I was a bit disappointed in the lack of discussion of the processes involved that led to this combination of events over Spring and Summer. Figure 7 provides some insight into how the unique the RHW event was but stronger statements could be made about whether the spring event was a necessary condition for the RHW.

4) The concluding paragraph seems to suggest that the positive GPP anomaly in spring offsets the negative anomaly in summer such that the net effect is a positive impact. This is slightly misleading given there were still substantial consequences on crop productivity in summer. This makes it hard to reconcile the 'GPP compensation' as nec-

essarily a positive impact. This text needs careful revising.

Minor Comments:

5) There are a couple of instances where the text is awkward and could be revised e.g. page 2 line 21: 'In 2010 the depleted state of soil moisture was one important driver which locally amplified the high temperature regime' could be written as 'In 2010 a negative soil moisture contributed to increased temperatures'

6) When calculating anomalies, it is still useful to know what they are anomalous to. Please include the reference period to which the anomalies are derived from for all figures that are showing anomalies.

7) I don't understand the phrase 'impact-agnostic approach' on Page 3

8) Page 3-4 "For instance, a popular approach is to consider an observation in a single (ideally normally distributed) anomaly variable to be extreme if it deviates by more then two standard deviations from the variable's mean values." Perhaps include references here that use this approach. Many studies on extremes also use other definitions from the Expert Team on Sector-specific Climate Indices (ET-SCI) which use percentile thresholds to identify extremes.

9) Page 4, line 11: replace 'constellations' with 'combinations'

10) Page 4, last paragraph: it may be useful to note the native resolution of the datasets that are used. I gather that the regridding of the land cover classification was done using a conservative or nearest neighbour approach?

11) Page 5, first paragraph: is there a reason why the median is used? Obviously because it is less susceptible to outliers but perhaps worth noting why. I'm also not sure who would define regional extremes using a global threshold so perhaps omit this suggestion and simplify the narrative.

12) Page 5, line 20: 'sort the median seasonal cycles according to the permutation of

temperature' I'm not sure what is meant by 'permutation of temperature'

13) It would be nice if Figure 4 and Figure B2 could be combined as this shows the contrast between the hydrometeorological and biospheric events and at the moment this feels concealed in the present form

14) Don't forget to do a spell check!

15) Page 9, second paragraph: I'm not quite comfortable with the phrase "In total, 41% of the summer carbon losses are compensated by an anomalously productive spring" because it implies that there was a recovery in GPP after the summer event which we don't actually know here. We only know that impact of the summer event is not as severe as it could have been because of the excess productivity in spring. Perhaps this can be resolved by using a word other than 'compensation'.

16) I like the narrative discussing the results according to vegetation type as this goes a long way to understanding differences in the spatiotemporal structure of the events.

17) The narrative for Figure 7 is too concise, here would be an opportunity to emphasise how unique the RHW compound event really was

18) Last sentence on page 13 seems to be contradictory to the narrative of the second paragraph on this page.

19) Page 14, line 3: 'constellation' makes me think of stars. I think 'conditions' would be more appropriate here.

20) Page 14, line 11: "this finding highlights the importance of forest ecosystems to mitigate the impacts of climate extremes" Be careful here, as there is some location dependence. Furthermore, how much is this a necessary result of the preconditioning in spring? The focus of the paper isn't the mitigation potential of forests so perhaps its better to remove this statement.

21) The text in supplementary section S1 seems to be repetition of the text in the main

manuscript. Either elaborate more or remove.

22) Supplementary Figure S3 4 – x axis labels: what is 'tempanoms' and how is this distinct from 'temp' – I'm guessing it's the anomaly? The caption needs more information to understand what is actually plotted here. Is the data aggregated to obtain the spatial mean or are all grid cells used to construct the linear models?

---

## Referee Comment (RC2) · Anonymous Referee #2 · 2 May 2018

Review for bg-2018-130, Flach et al., "Contrasting biosphere responses to hydrometeorological extremes: revisiting the 2010 western Russian Heatwave."

Flach and colleagues, using a multivariate spatiotemporal anomaly detection algorithm on both climate and ecosystem variables, assess the response of productivity to the Russian heat wave of 2010. Motivated by the potential for inconsistencies in the climate event and the biospheric impact (which they suggest is a function of disciplinary divides) they find that an anomalous spring warming event in both the biosphere and climate increased GPP prior to the actual heat wave itself, which occurred later in summer, thus offsetting the negative productivity effects. They note that the compensation

occurs in different ecosystems–losses dominated in lower latitude managed ecosystems, such as crop land, while spring gains dominated in higher latitude forested regions. During the heat event itself, they attributed the differential response of forests and crops to different water management strategies of the vegetation classes. Overall the paper is a nice contribution and appears methodologically sound (if not a bit overcomplicated in places). I have a few comments and suggestions for the authors to consider that I hope will help improve the clarity and argument of the paper.

Main comments:

1. Stated motivation: While I am sympathetic to the larger issue that climate extremes and climate impacts are distinct domains and that extremes may not necessarily map to impacts, I find parts of the introduction to be somewhat of a 'straw man.' The hydro and bio perspectives generally do agree on the Russian heat wave–warm temperatures, along with dry soils leads to carbon loss. Consider the fact, for example, that the authors' very own agnostic algorithm finds the same two events in both the met and bio fields; it suggests that the RHW at least, this disconnect does not lead to inconsistent interpretations or conclusions among different disciplines. The notion that there isn't a one-to-one mapping between the geophysical event and the biophysical impact is certainly important for accurately representing the total effects as a function of the differential vulnerabilities of ecosystems. The authors rightfully emphasize this. However, the notion that this issue is emblematic of some kind of disciplinary divide is over-reach, or at the very least, is not supported by the literature the authors cite here. I heartedly agree that a call for an integrative perspective is a good one, as it can provide both a richer treatment of an extreme event and a basis for better impacts prediction, but the way the introduction is cast at present overstates the extent to which disciplinary perspectives are or were an issue in some kind of misdiagnosis of the RHW. This can be seen, for example, at 3.10, where the authors state that because the GPP declines were not as large as the temperature anomalies in Fig. 1, that this is somehow reflective of "different disciplinary perspectives" rather than of the complexity of the Earth

system itself. . .leading the authors to "suspect [. . .it] might become an issue in studies of this kind." If the authors provided a stronger basis in the literature of inconsistent conclusions of the impacts of the RHW or similar events based on disciplinary divides, then sure, the way the intro is written can stand, but I think as is, it overstates it as a problem and diminishes the scientific conclusions of the paper, which are interesting in and of themselves. The point is, those interesting results and the science itself, gets a bit lost in the straw man discourse. Edits to the text can fix this.

2. Two events v. one event: My comment here is a corollary to the above about how the paper is cast relative to the literature. The authors are taking two separate events in 2010, an anomalous spring and an anomalous summer, and integrating the impacts across those two events and casting it as the net effects of the RHW, rather than simply examining the net consequence of the RHW itself. Certainly the spring event is crucial to providing a picture of GPP over the growing season and this approach makes sense for the effects of the full growing season on GPP: the extent to which the spring anomaly primed, compensated, or otherwise interacted with the RHW is important. But conceptually the authors need to make clear that simply combining them does not constitute the carbon response to the RHW, for as written, the RHW impacts are presented as the net effects of two separate events, rather than just the heat wave. Given the motivation the authors lead with (i.e., that there is an inherent potential for some kind of mismatch from the atmosphere down and the biosphere up), calling the impact of the RHW the integration of two distinct events seems like an issue. Perhaps the results should be recast around the compensation effects of spring growth on total growing season GPP in the year of the Russian heat wave. I think just making this distinction clearer is important. The net impact of the RHW is not growing season GPP, which includes the anomalous spring, it's just the GPP loss during the RHW. These integrations can be seen in Tables 1 and 2, S4.1, etc. Further complicating this is the fact that the actual losses and gains of GPP are domain integrated, and the domain integration is a function of the detection algorithm. Certainly the authors discuss that the compensation occurs in a fundamentally different part of the domain and land cover

class than the heat wave impacts, so I find the combination a bit misleading–it occurs in a different location and time than the actual heat wave–1TgC in crops is fundamentally different than that for forests (though from a carbon accounting perspective perhaps not). This again, is just about how the results are presented, particularly the res+/res-, not the results themselves.

3. Merits of the detection approach: Part of the basis of this manuscript is that a much more sophisticated detection approach is needed to accurately represent the biophysical impacts of climate extremes. If one simply did the detection–as is typical– at the grid point scale on the hydrometeorological fields and then composited on the biophysical fields for the same dates as the meteorological anomaly, would the results and/or conclusions substantially differ? At 5.10 the authors claim that for a short time series a traditional threshold approach would be problematic. Is there evidence for this? The authors still have to perform a sensitivity analysis of their results to the chosen threshold (S4.1). At some places the paper feels needlessly complex–perhaps the authors could better justify their complicated analytical choices?

4. Model of factors explaining the GPP response. This section (S3), which is referred to in the main, but relegated to the Supplemental could be better emphasized and explained. For example, the factors in the hierarchical modeling approach are not independent. Are interaction variables used to address this issue? Given the confounding of latitude and temperature and land cover class, why not add latitude to the regression hierarchy to see its explanatory power, given the sentiment at 12.3?

5. Attribution to uWUE differences. The authors attribute the reduced GPP declines during the summer event of forests in part due to the uWUE. Certainly this has a role to play. One could also imagine uWUE being an explanatory variable in the model presented in section S3 as well–could the authors add that? It seems like the authors are positioned to better attribute whether it was the absolute magnitude of the temperature itself (which diminished as a function of latitude) or something innate to the land cover classes (and their underlying WUE), which just so happens to vary as a function of

latitude. The model seems like an ideal place to disassociate these factors. Regarding the spring event and soil moisture depletion carry-over effects under forcing discussed at 12.22-13.6, Mankin et al., Journal of Climate 2017 and Mankin et al. GRL 2018 note that increased productivity is associated with such carry over effects in some of the models, regionally and globally under forcing.

Minor comments:

11.5: I don't understand the soil moisture in Fig. 7. Is it the normalized measure? Is it the m3/m3? Can the authors add contours if the forests separate by latitude in 7b?

Grammar/spelling throughout could be improved.

1.16: (e.g., a vegetation index) inconsistency in comma usage after e.g. and i.e.

2.29: not sure the name is "heat summer"

2.32: a, not an, hydrometeorological

5.8: grammar ("in high")

5.4: Why not leave them as missing data?

21.32 "spatiotemporal" not "…temporla"

Author contributions: "wrote" not "ote"

---

## Author Comment (AC1) · 7 Jun 2018

*Response* to Anonymous Referee #1

Summary:

This manuscript presents a case study analysis to examine the impacts of compound events through a comparison of hydrological (via soil moisture) and biospheric (via GPP) perspectives in the season preceding, and during, the Russian 2010 heatwave. The paper provides a case for why singular extreme events need to be examined under different perspectives to understand the full implications of these events across multiple sectors. It is a nice study however I was anticipating a more indepth analysis of the processes that connect the two events. Its almost there and perhaps only requires minor revision of the text to achieve this.

*Response: We would like to thank the reviewer for the positive evaluation of our manuscript and agree that the discussion regarding the processes connecting the hydrometeorological and biopsheric event, and the connections between the spring and summer events can be substantially improved. We do our very best to provide a more in-depth discussion which hopefully addresses the reviewer's concerns. Specifically, we will add a paragraph to the introduction (see reply to 1), and the discussion (see reply to 3).*

Main Comments:

1) The hydrological event and the biospheric events don't have the same spatial coverage which makes it hard for those new to the concept of compound events to appreciate how the events evaluated in the manuscript are indeed related. Could the authors perhaps provide a stronger case for why these distinctive events should be considered together beyond the 'different disciplinary perspectives' by delving into how one may be a result of the other. The commentary around Figure 1 on page 3 makes it difficult to reconcile the fact that the two events are related. Perhaps part of the confusion also stems from having a spring event, a summer event and then considering these events defined in terms of either the biospheric and hydrological perspective (so effectively giving 4 events to compare). I think this can be resolved by amending the text and including more discussion on how these events fit together.

*Response: We highly appreciate the reviewers' perspective on compound events. We already elaborate a little bit on the biospheric response to heatwaves and droughts (p. 2, l.25-31), but we agree with the reviewer, that the link between biosphere and atmosphere, as well as spring and summer is not well explained. Thus, we will extend the commentary around Figure 1 and elaborate on connections between hydrometeorology and biosphere as well as spring and summer (p.3, l.5) as follows:*

*Temperature anomalies exceeded more than 10~K in both spring and summer, but they lead to distinctive anomalies in gross primary productivity (GPP). Positive GPP anomalies occurred during the spring event, whereas negative GPP anomalies are occurring during the summer heatwave. The positive GPP response in spring might be a reaction to warmer, more optimal spring temperatures (Wang et al, 2017) possibly accompanied by enough water availability. However, negative GPP anomalies in summer occurred only in areas south of 55°N (Fig. 1c) indicating that the GPP response involves much more processes than high temperatures and drought during the unique RHW. As already indicated by Smith, 2011, the connection between biosphere and hydrometeorology is much more complex than just a direct one-to-one mapping. Further complicating this issue is the fact that the summer event cannot be investigated without the previous spring, as both seasons are inherently related via memory effects in water availability. Increased GPP in spring may negatively influence soil moisture and thus GPP during summer (Buermann et al., 2013). In Summary, comparing ...*

2) The narrative in section 2.2 was hard to follow in that there is some information that may be better to remove (e.g. defining extremes using global thresholds) or a dependence on jargon that not everyone may understand (some examples noted in the minor comments). Given that the manuscript aims to articulate a methodology for extracting information on compound events this could be revised. Would it be possible to add some illustration to the schematic in Figure 2 to clarify how the spatiotemporal segments are defined and extracted.

*Response: We agree with the reviewer, that section 2.2 can be improved. We will completely revise the section. We will remove unecessary parts (e.g. the global thresholds) and avoid jargon whenever possible. We will add the following schematic Figure to illustrate the extraction of the spatial segments.*

[Figure]

3) I was a bit disappointed in the lack of discussion of the processes involved that led to this combination of events over Spring and Summer. Figure 7 provides some insight into how the unique the RHW event was but stronger statements could be made about whether the spring event was a necessary condition for the RHW.

*Response: We agree with the reviewer, that it would indeed be very nice to show the connection between summer and spring events and whether this kind of unique summer events only happen preconditioned on an anomalous spring. However, this would require running process based model simulations (as some of the coauthors already did for evaluating the general presence of spring summer compensation effects in Sippel et al. 2017) which goes beyond the scope of this paper - focussing more on a statistical detection. We agree that this is a very relevant question that can be addressed in a follow up study.*

*To address the reviewer's need for process based connections between the spring and summer events we suggest to intensify the discussion about the biophysical processes that could link spring and summer anomalies. Several works suggest that spring warming leads to depleted soil moisture in summer, thus amplifying the summer droughts (e.g., Buermann et al., 2013, ERL, Wolf et al., 2016, PNAS). To address this issue, we will add a paragraph to the introduction (see reply to 1), and we will add a paragraph before p.12, l. 22 – p. 13, l. 10. with a more in-depth discussion as folows:*

*Another important aspect is that the combination of the anomalous spring and the unique heatwave in summer might be inherently connected via land surface feedbacks. Buermann et al., 2013 showed that warmer springs going in hand with earlier vegetation activity negatively affect soil moisture in summer. It is a general observation that warm and dry springs enhance summer temperatures during droughts, which suggests the presence of soil-moisture temperature feedbacks across seasons (Haslinger et al., 2017). In case of the Russian heatwave 2010, soil moisture was one of the main drivers (Hauser et al., 2016), in hand with persistent atmospheric pressure patterns (Miralles et al., 2014). Thus, we suspect that the spring event is connected to the summer heatwave in 2010, if not setting the preconditions for a heatwave of this unique magnitude.*

4) The concluding paragraph seems to suggest that the positive GPP anomaly in spring offsets the negative anomaly in summer such that the net effect is a positive impact. This is slightly misleading given there were still substantial consequences on crop productivity in summer. This makes it hard to reconcile the 'GPP compensation' as necessarily a positive impact. This text needs careful revising.

*Response: We would like to thank the reviewer for pointing us that the concluding paragraph could be misunderstood. Our intention was not to suggest that the integrated net effect of the events in Russia 2010 was a positive one in terms of carbon budget and tried our best to avoid this kind of misunderstanding, e.g. state that in the first part of the concluding sentence (p.14, l. 16): "Although the integrated impact on gross primary production of the hydrometeorological conditions is strongly negative, it is important to notice the strong compensatory effects due to differently affected ecosystem types, as well as duration and timing of the extreme events." We will replace „strong" with „partial" to avoid missunderstandings.*
*To prevent further misunderstanding, we will exchange "compensate" with "partly compensate" or "compensation" with "partial compensation" in the conclusions, and the abstract. Furthermore, we will add a sentence on p. 14, l.11 clarifying this once more: "Please note, that the integrated impact of the 2010 events on the carbon balance is strongly negative."*

 Minor Comments:

5) There are a couple of instances where the text is awkward and could be revised e.g. page 2 line

21: 'In 2010 the depleted state of soil moisture was one important driver which locally amplified the high temperature regime' could be written as 'In 2010 a negative soil moisture contributed to increased temperatures'

*Response: We thank the reviewer and we will change it accordingly and go once again through the text to find such awkward instances.*

6) When calculating anomalies, it is still useful to know what they are anomalous to. Please include the reference period to which the anomalies are derived from for all figures that are showing anomalies.

*Response: We agree with the reviewer and add this information as suggested to Fig. 1, 4, 6, 8, A1, B1.*

7) I don't understand the phrase 'impact-agnostic approach' on Page 3

*Response: "Impact-agnostic" may be just our own jargon. We meant here, that our approach is independent, whether the event is related to a positive or a negative impact. We will remove the phrase.*

8) Page 3-4 "For instance, a popular approach is to consider an observation in a single (ideally normally distributed) anomaly variable to be extreme if it deviates by more then two standard deviations from the variable's mean values." Perhaps include references here that use this approach. Many studies on extremes also use other definitions from the Expert Team on Sector-specific Climate Indices (ET-SCI) which use percentile thresholds to identify extremes.

*Response: We will include references as suggested by the reviewer.*

9) Page 4, line 11: replace 'constellations' with 'combinations'

*Response: We will replace it.*

10) Page 4, last paragraph: it may be useful to note the native resolution of the datasets that are used. I gather that the regridding of the land cover classification was done using a conservative or nearest neighbour approach?

*Response: We thank the reviewer for the suggestion. The spatial resolution of the original data-*

*sets will be provided. Regridding the land cover classification (original: 300m) was done by using the major land cover class for the new resolution. We will add this information accordingly.*

11) Page 5, first paragraph: is there a reason why the median is used? Obviously because it is less susceptible to outliers but perhaps worth noting why. I'm also not sure who would define regional extremes using a global threshold so perhaps omit this suggestion and simplify the narrative.

*Response: We thank the reviewer for this important comment: Yes, we used the median because it is less susceptible to outliers. We will add this explanation (p.5, l.3) and remove the part about global thresholds (p.5, l.6-9).*

12) Page 5, line 20: 'sort the median seasonal cycles according to the permutation of temperature' I'm not sure what is meant by 'permutation of temperature'

*Response: We thank the reviewer for pointing to this jargon issue. We meant that the seasonal cycle of temperature is sorted (e.g. high to low). We memorize the order (permutation) and apply the same ordering to the other seasonal cycles. We will change the text to explain exactly what we did.*

13) It would be nice if Figure 4 and Figure B2 could be combined as this shows the contrast between the hydrometeorological and biospheric events and at the moment this feels concealed in the present form

*Response: We will combine them as new Figure 4.*

14) Don't forget to do a spell check!

*Response: We will go through the text once again. We also highly appreciate that Biogeosciences now performs a carful language check previous to publication.*

15) Page 9, second paragraph: I'm not quite comfortable with the phrase "In total, 41% of the summer carbon losses are compensated by an anomalously productive spring" because it implies that there was a recovery in GPP after the summer event which we don't actually know here. We only know that impact of the summer event is not as severe as it could have been because of the

excess productivity in spring. Perhaps this can be resolved by using a word other than 'compensation'.

*Response: We thank the reviewer pointing to the potential misunderstanding regarding the "compenstion" effect and to the relevance of recovery after the heatwave. We checked for "extreme" GPP anomalies after the summer event, but we could not find any. Thus, vegetation might still be slightly less productive than the years before and after, but it is still considered to be within "normal" variability by the detection approach. This suggests that the effect of the heatwave is limited in time, and that ecosystems are able to recover relatively quickly. We will add a sentence of post-heatwave recovery in the manuscript on p.9, l.6.*

*Regarding the reviewer's concerns about the "compensation" effect we will rewrite the sentence to: "If we consider the annually-integrated effect of the spring and summer anomalies, spring carbon gains are estimated to offset 41% of the subsequent carbon losses in summer." In other cases, we would like to stick to the term "compensation" because it is already coined by previous literature on this topic (e.g. Wolf et al., 2016,; Sippel et al., 2017).*

16) I like the narrative discussing the results according to vegetation type as this goes a long way to understanding differences in the spatiotemporal structure of the events.

*Response: We would like to thank the reviewer for this positive feedback.*

17) The narrative for Figure 7 is too concise, here would be an opportunity to emphasise how unique the RHW compound event really was

*Response: We will add a few sentences on that.*

18) Last sentence on page 13 seems to be contradictory to the narrative of the second paragraph on this page.

*Response: We thank the reviewer for pointing to this issue. We will make clear in the beginning of the second paragraph that the compensation effects mentioned there are more general and not directly related to the case study of the Russian heatwave (p.13, l.12): "They show that in general warm springs increasingly compensate summer productivity losses in Europe, ..."*

*Furthermore we will emphasize that the last sentence on p.13 is only related to the RHW: "Regarding the RHW in particular, compensation effects remain unconsidered in previous studies to the best of our knowledge".*

19) Page 14, line 3: 'constellation' makes me think of stars. I think 'conditions' would be more appropriate here.

*Response: We will change it as suggested.*

20) Page 14, line 11: "this finding highlights the importance of forest ecosystems to mitigate the impacts of climate extremes" Be careful here, as there is some location dependence. Furthermore, how much is this a necessary result of the preconditioning in spring? The focus of the paper isn't the mitigation potential of forests so perhaps its better to remove this statement.

*Response: We will remove the statement.*

21) The text in supplementary section S1 seems to be repetition of the text in the main manuscript. Either elaborate more or remove.

*Response: We will remove it from the supplementary and merge the information into the revised paragraph 2.2 (spatiotemporal segmentation).*

22) Supplementary Figure S3 4 – x axis labels: what is 'tempanoms' and how is this distinct from 'temp' – I'm guessing it's the anomaly? The caption needs more information to understand what is actually plotted here. Is the data aggregated to obtain the spatial mean or are all grid cells used to construct the linear models?

*Response: We apologize for the bad labeling of Figure S3 4. We will change it in T anomalies and the other abbreviations accordingly. We will also add more information about the section in the main manuscript (as a request from reviewer#2) and revise the paragraph at S3, add explainations about the methods to the text, and add information to the caption. Regarding the reviewers question on the aggregation: All grid cells are used to construct the linear models without aggregation.*

---

## Author Comment (AC2) · 7 Jun 2018

*Response to* Anonymous Referee #2

Review for bg-2018-130, Flach et al., "Contrasting biosphere responses to hydrometeorological extremes: revisiting the 2010 western Russian Heatwave."

Flach and colleagues, using a multivariate spatiotemporal anomaly detection algorithm on both climate and ecosystem variables, assess the response of productivity to the Russian heat wave of 2010. Motivated by the potential for inconsistencies in the climate event and the biospheric impact (which they suggest is a function of disciplinary divides) they find that an anomalous spring warming event in both the biosphere and climate increased GPP prior to the actual heat wave itself, which occurred later in summer, thus offsetting the negative productivity effects. They note that the compensation occurs in different ecosystems–losses dominated in lower latitude managed ecosytems, such as crop land, while spring gains dominated in higher latitude forested regions. During the heat event itself, they attributed the differential response of forests and crops to different water management strategies of the vegetation classes. Overall the paper is a nice contribution and appears methodologically sound (if not a bit overcomplicated in places). I have a few comments and suggestions for the authors to consider that I hope will help improve the clarity and argument of the paper.

*Response: We would like to thank the reviewer for the positive evaluation.*

Main comments:

1. Stated motivation: While I am sympathetic to the larger issue that climate extremes and climate impacts are distinct domains and that extremes may not necessarily map to impacts, I find parts of the introduction to be somewhat of a 'straw man.' The hydro and bio perspectives generally do agree on the Russian heat wave–warm temperatures, along with dry soils leads to carbon loss. Consider the fact, for example, that the authors' very own agnostic algorithm finds the same two events in both the met and bio fields; it suggests that the RHW at least, this disconnect does not lead to inconsistent interpretations or conclusions among different disciplines. The notion that there isn't a one-to-one mapping between the geophysical event and the biophysical impact is certainly important for accurately representing the total effects as a

function of the differential vulnerabilities of ecosystems. The authors rightfully emphasize this. However, the notion that this issue is emblematic of some kind of disciplinary divide is over-reach, or at the very least, is not supported by the literature the authors cite here. I heartedly agree that a call for an integrative perspective is a good one, as it can provide both a richer treatment of an extreme event and a basis for better impacts prediction, but the way the introduction is cast at present overstates the extent to which disciplinary perspectives are or were an issue in some kind of misdiagnosis of the RHW. This can be seen, for example, at 3.10, where the authors state that because the GPP declines were not as large as the temperature anomalies in Fig. 1, that this is somehow reflective of "different disciplinary perspectives" rather than of the complexity of the Earth system itself. . .leading the authors to "suspect [. . .it] might become an issue in studies of this kind." If the authors provided a stronger basis in the literature of inconsistent conclusions of the impacts of the RHW or similar events based on disciplinary divides, then sure, the way the intro is written can stand, but I think as is, it overstates it as a problem and diminishes the scientific conclusions of the paper, which are interesting in and of themselves. The point is, those interesting results and the science itself, gets a bit lost in the straw man discourse. Edits to the text can fix this.

*Response: We would like to thank the reviewer for the positive view on the scientific conclusions of our manuscript. We agree with the reviewer, that we somehow overstated the disciplinary differences on the existing literature at the basis of the Russian Heatwave. We will carefully revise the abstract and the introduction to fix issues of this kind. In particular, we will rephrase the motivation at 3.10. along the lines the reviewer suggested (more focused on our own results, highlighting the call for an integrated perspective):*

*"The objective of this paper is therefore to revisit the RHW and to investigate the GPP response during the spring event and the summer heatwave when adopting a hydrometeological driver vs. a biospheric perpective."*

*Furthermore, we will reformulate the sentence on 3.2-3 to: "However, an integrated assessment including the hydrometeorological and the biospheric domain may facilitate our knowledge about the RHW. In particular, we highlight one aspect of the RHW which can easily be seen, i.e., if we look at the zonal evolution of the RHW in both domains" and remove two sentences in the abstract (1.5 and 1.16-17).*

2. Two events v. one event: My comment here is a corollary to the above about how the paper is cast relative to the literature. The authors are taking two separate events in 2010, an anomalous spring and an anomalous summer, and integrating the impacts across those two events and casting it as the net effects of the RHW, rather than simply examining the net consequence of the RHW itself. Certainly the spring event is crucial to providing a picture of GPP over the growing season and this approach makes sense for the effects of the full growing season on GPP: the extent to which the spring anomaly primed, compensated, or otherwise interacted with the RHW is important. But conceptually the authors need to make clear that simply combining them does not constitute the carbon response to the RHW, for as written, the RHW impacts are presented as the net effects of two separate events, rather than just the heat wave. Given the motivation the authors lead with (i.e., that there is an inherent potential for some kind of mismatch from the atmosphere down and the biosphere up), calling the impact of the RHW the integration of two distinct events seems like an issue. Perhaps the results should be recast around the compensation effects of spring growth on total growing season GPP in the year of the Russian heat wave. I think just making this distinction clearer is important. The net impact of the RHW is not growing season GPP, which includes the anomalous spring, it's just the GPP loss during the RHW. These integrations can be seen in Tables 1 and 2, S4.1, etc. Further complicating this is the fact that the actual losses and gains of GPP are domain integrated, and the domain integration is a function of the detection algorithm. Certainly the authors discuss that the compensation occurs in a fundamentally different part of the domain and land cover class than the heat wave impacts, so I find the combination a bit misleading–it occurs in a different location and time than the actual heat wave–1TgC in crops is fundamentally different than that for forests (though from a carbon accounting perspective perhaps not). This again, is just about how the results are presented, particularly the res+/res-, not the results themselves.

*Response: We would like to thank the reviewer for this comment. We apologize if the net effects of the Russian Heatwave (RHW) can be misunderstood as integrated spring and summer effect. We will carefully revise the manuscript to address this issue. Furthermore, we did not mention in the manuscript, that there is no event after summer. Thus the annual integration over the events in the growing season in 2010 equals the integration over spring and summer. We add a sentence to clarify this issue on p.9, l.6: "Please note, that we did not find extreme events after summer, which implies a fast recovery of vegetations activity after summer. Integrations over the spring*

*and summer events thus equals the annual integration.“ Furthermore, we will reformulate
"integrated over spring and summer“ to "annually integrated“.*

*Reviewer #1 expressed concerns along the same lines, particularly with respect to process based
connections between the spring and summer event. We will provide a more in-depth discussion
abouth how the spring and summer event might be related:*

*First, we will add a paragraph to the introduction (p.5, l.3): "Temperature anomalies exceeded
more than 10~K in both spring and summer, but they lead to distinctive anomalies in gross
primary productivity (GPP). Positive GPP anomalies occurred during the spring event, whereas
negative GPP anomalies are occurring during the summer heatwave. The positive GPP response
in spring might be a reaction to warmer, more optimal spring temperatures (Wang et al, 2017)
possibly accompanied by enough water availability. However, negative GPP anomalies in
summer occurred only in areas south of 55°N (Fig. 1c) indicating that the GPP response involves
much more processes than high temperatures and drought during the unique RHW. As already
indicated by Smith, 2011, the connection between biosphere and hydrometeorology is much more
complex than just a direct one-to-one mapping. Further complicating this issue is the fact that the
summer event cannot be investigated without the previous spring, as both seasons are inherently
related via memory effects in water availability. Increased GPP in spring may negatively
influence soil moisture and thus GPP during summer (Buermann et al., 2013). In Summary,
comparing ... "*

*Second, we will add a paragraph to the discussion p.12, l. 22 – p. 13, l. 10. as folows: "Another
important aspect is that the combination of the anomalous spring and the unique heatwave in
summer might be inherently connected via land surface feedbacks. Buermann et al., 2013 showed
that warmer springs going in hand with earlier vegetation activity negatively affect soil moisture
in summer. It is a general observation that warm and dry springs enhance summer temperatures
during droughts, which suggests the presence of soil-moisture temperature feedbacks across
seasons (Haslinger et al., 2017). In case of the Russian heatwave 2010, soil moisture was one of
the main drivers (Hauser et al., 2016), in hand with persistent atmospheric pressure patterns
(Miralles et al., 2014). Thus, we suspect that the spring event is connected to the summer
heatwave in 2010, if not setting the preconditions for a heatwave of this unique magnitude. "*

3. Merits of the detection approach: Part of the basis of this manuscript is that a much more
sophisticated detection approach is needed to accurately represent the biophysical impacts of

climate extremes. If one simply did the detection–as is typical– at the grid point scale on the hydrometeorological fields and then composited on the biophysical fields for the same dates as the meteorological anomaly, would the results and/or conclusions substantially differ? At 5.10 the authors claim that for a short time series a traditional threshold approach would be problematic. Is there evidence for this? The authors still have to perform a sensitivity analysis of their results to the chosen threshold (S4.1). At some places the paper feels needlessly complex– perhaps the authors could better justify their complicated analytical choices?

*Response: We would like to thank the reviewer for the critical analysis of our detection approach. The main advantage of our multivariate detection approach is that we can integrate information about several variables simultaneously and might also detect rare combinations of variables which are not detected as extreme individually (4.9-14). However, the events in Russia 2010 are not an example of this kind. Thus, we agree with the reviewer, that it is possible to get similar results by combining several univariate detection approaches for the Russian heatwave. Combining univariate detection approaches would require to choose a threshold for each variable individually. Performing a full sensitivity analysis of the choosen thresholds would lead to a combination of many possible thresholds which would render high dimensional unfeasable.*

*We would like to thank the reviewer pointing us to our claim at 5.10. which might be suspect to missunderstanding. Our intention was not to state that the traditional threshold approach itself is problematic. We wanted to state that the underlying assumption (equal distribution of extreme events among all grid cells) is most likely not met for short time series (here: 11 years). A 10% threshold in a 10 year time series would seelct one extreme year in each grid cell (not more, not less). There are regions where extremes events occur more often or are longer than one year, e.g., California (Griffin, D., and K. J. Anchukaitis, 2014 , GRL) or where by chance no extreme event at all is occuring in the given time frame. Our spatiotemporal segmentation is addressing this issue by choosing thresholds over larger areas with compareable climate and phenology. As a request also from Reviewer#1 we will completely revise the section 2.2 including a new schematic figure for the spatiotemporal extraction. In this process, we will rephrase the given part above, which is suspect to missunderstanding. Furthermore, we will justify complicated analytical choices as suggested by the reviewer, remove unneccessary parts (global thresholds, local thresholds), merge it with the information in S1, and avoid jargon whenever possible.*

4. Model of factors explaining the GPP response. This section (S3), which is referred to in the main, but relegated to the Supplemental could be better emphasized and explained. For example, the factors in the hierarchical modeling approach are not independent. Are interaction variables used to address this issue? Given the confounding of latitude and temperature and land cover class, why not add latitude to the regression hierarchy to see its explanatory power, given the sentiment at 12.3?

*Response: We would like to thank the reviewer for his interest in the Section S3 and agree that the section can be much better emphasized and explained. We will carefully revise the section and introduce more information in the main manuscript on 12.1. The factors in the hierarchical modeling approach are indeed not independent. However, the hierarchical partitioning after Chevan and Sutherland (1991), is exactly made for this kind of issues. The method extracts the independent contribution of interacting variables.*

*We would also like to thank the reviewer for the idea to extent the regression model with the factor latitude. Indeed, latitude has a very high independent explanatory power (Figure below) which is comparable to the importance of land cover type in summertime. The high independent explanatory power indicates that latitude provides additional information, which is not already contained in the other factors (e.g., land cover type or absolute temperatures). In particular access to deeper water (and soil type) might be factors not contained in the model, but also changing with latitude and therefore possibly explaining the importance of latitude. Apart from including more information about the method, we will include more information on S3 in the main manuscript and the new results in section 3.3 with the following paragraph and reformulate other sections if necessary:*

*"To disentangle the variable importance of the different confounding factors, we run a simple linear regression model which tries to explain GPP as function of the hydrometeorological driver variables (temperature, precipitation, radiation, surface moisture, anomalies and absolute values), as well as vegetation type, duration and latitude (Supplementary S3). We use an algorithm after \citet{Chevan:1991wg} which extracts the independent contribution of the variable importance related to this particular variable regardless of the model complexity or dependencies among variables.*

*The model reveals from a statistical point of view, that vegetation type and the latitudinal gradient are the most important variables explaining GPP during the summer event, followed by the hydrometeorological drivers. Access to deeper water and soil type as well as non-linear feedbacks are factors which are not represented in the model, but might explain the high variable importance of latitude."*

[Figure]

5. Attribution to uWUE differences. The authors attribute the reduced GPP declines during the summer event of forests in part due to the uWUE. Certainly this has a role to play. One could also imagine uWUE being an explanatory variable in the model presented in section S3 as well–could the authors add that? It seems like the authors are positioned to better attribute whether it was the absolute magnitude of the temperature itself (which diminished as a function of latitude) or something innate to the land cover classes (and their underlying WUE), which just so happens to vary as a function of latitude. The model seems like an ideal place to disassociate these factors.

*Response: In general we like the idea to add uWUE as a explanatory variable in the model. However, uWUE is defined as GPP * VPD^0.5 / ET. Thus adding uWUE as factor to the model*

*would be somehow circular, as the target variable (GPP) is contained in the possible factor uWUE. Thus, we think adding uWUE would be inappropriate from a statistical point of view.*

Regarding the spring event and soil moisture depletion carry-over effects under forcing discussed at 12.22-13.6, Mankin et al., Journal of Climate 2017 and Mankin et al. GRL 2018 note that increased productivity is associated with such carry over effects in some of the models, regionally and globally under forcing.

*We would like to thank the reviewer for the two additional references which we found very interesting. We will add the references to the discussion about soil moisture carry over effects.*

Minor comments:

11.5: I don't understand the soil moisture in Fig. 7. Is it the normalized measure? Is it the m3/m3? Can the authors add contours if the forests separate by latitude in 7b?

*Response: It is m3/m3.*

Grammar/spelling throughout could be improved.

*Response: We will go through the text once again. We also highly appreciate that Biogeosciences now performs a carful language check previous to publication.*

1.16: (e.g., a vegetation index)

*Response: The sentence will be removed to address major comment 1)*

inconsistency in comma usage after e.g. and i.e.

*Response: Will be checked for consistency*

2.29: not sure the name is "heat summer"

*Response: changed into "European heatwave 2003"*

2.32: a, not an hydrometeorological

*Response: Will be changed*

5.8: grammar ("in high")

*Response: Will be corrected*

5.4: Why not leave them as missing data?

*Response: We would like to thank the reviewer for this comment. Indeed, it would be an option to leave the data as missing in case all variables are missing at one observation, excluding the observation from the multivariate detection. However, in comparison to univariate event detection, our multivariate algorithm requires that all variables are available. Thus, there are many more missing instances, i.e., cases which have only one of the variables missing, all others are available. We will add the following sentence on that:*

*"The gap filling is necessary for a multivariate detection approach as there are many more cases in which one variable is missing in the multivariate cube compared to a univariate data stream."*

21.32 "spatiotemporal" not ". . .temporla"

*Response: Will be corrected*

Author contributions: "wrote" not "ote"

*Response: Will be corrected*

---

## Author Response (AR2)

*Dear Dr. Paul Stoy,*

*Please find a point by point response to the referee comments attached.*

*Sincerely,*
*Milan Flach (on behalf of all co-authors)*

Review of the Revised Manuscript: "Contrasting biosphere responses to hydrometerological extremes: revisiting the 2010 western Russian Heatwave" by Flach and coauthors

Main Comments

As was raised in my first review of this manuscript, what I find hardest to reconcile is the concept of GPP compensation between two ecosystems that are spatially distinct and that occur at different times.

*Response: We are grateful to the reviewer raising this important point again and take his critique about the concept of GPP compensation very seriously. In line with Wolf et al., 2016, PNAS, and Sippel et al., 2017, ERL, we think that compensation as used here means just highlighting the positive and negative contribution to a summary statistics, which is aggregation in this case. We think that this is valid spatially as well as temporally:*

*1) Spatial aggregation: It is important to note that we do not aggregate between two spatially distinct hydrometeorological extreme events. It is the very same hydrometeorological extreme event, which drives different responses in different ecosystems. Furthermore, these two ecosystems might have distinct centroids (500km) of their peak response to the very same heatwave, but they are also spatially adjacent as they share a common boundary over roughly more than 3300km (30°E-60°E without taking small variations into account, Fig, 7a). The response of both adjacent ecosystems hit by the same extreme event is nevertheless different (Fig, 7b).*

*2) Different times: We aggree with the reviewer that just aggregating over spring and summer as was written in the former version of the manuscript is special. However, the aggregation over spring and summer equals the annual aggregation or the aggregation over the grwoing season in our case as we did not find anomalies in autumn or winter. We mention it in the text at p.10, l.10 and changed spring and summer in annually for better understanding. Additionally, the connection (and thus, aggregation) between spring and summer events makes sense from a process-based view due to soil moisture carry-over effects. Plants use water for additional productivity during warm springs, which is lacking in the following summer. We elaborate on that in the introduction at p.3, l.15-16 and the discussion on p.14, l.6, and p. 14, l.17-23.*

When aggregating over the spring and summer corresponding to the RHW the text makes it seem like overall the impact of the RHW was 'good' in terms of the net change in GPP integrated over the whole region.

*Response: We carefully checked the text for instances leaving this impression. We apologize for the instances where this impression might have arisen. We addressed this point by rephrasing parts of the discussion (3rd and 4th paragraph) and the last paragraph of the conclusions.*

However it was the agricultural ecosystems that were severely impact over the summer. Although there is one remark that post-RHW GPP anomalies were negligible indicating a recovery of the agricultural ecosystems this could be emphasised more by the discussion of Figure 9.

*Response: We added another sentence on the post heatwave recovery by the discussion of Figure 9 (p.12, l.5).*

Despite the authors reassuring in the response to reviewers that they would carefully check the manuscript text I was very disappointed at how poorly this was done. I have now noted in the minor comments most of the instances that I have found in the hope that they can be rectified in the next revision.

*Response: We are grateful to the reviewer for checking the manuscript text and corrected the instances, which were noted in the minor comments, as well as further cases.*

Minor Comments

• The authors need to get someone to check the text more carefully. There are several instances where the tense changes from past to present which is quite irritating. e.g. "positive GPP anomalies occurred during the spring event, whereas negative GPP anomalies are occurring during the summer heatwave" It is not sufficient to note appreciation to the journal for language editing services if they have yet to be used.

*Response: We apologize for tense changes, which were overseen during the last corrections. We corrected the instances, which occurred especially during the 2nd paragraph of the introduction (dealing with the Russian heatwave event in the past). We had a language check by a half-native American, who was OK with the quality of the text.*

• Furthermore, when citing other research terms/phrases such as "probably breaking temperature records of several centuries" and "The RHW is often associated with an atmospheric blocking situation" are ambiguous. Did the RHW break records or not? Was the RHW associated with atmospheric blocking? (yes) Be specific.

*Response: We rephrased the above mentioned sentences to be more specific.*

• Replace: "a negative soil moisture contributed" with "a negative soil moisture anomaly contributed". Again, please check the text; one cannot assume that every reader will necessarily be able to know which words are missing.

*Response: We added the missing word and went through the text carefully again.*

• I would suggest replacing 3.5 "may facilitate our knowledge about the RHW" with "may further our understanding of the RHW"

*Response: Done.*

• 3.10: "The positive GPP response in spring might be a reaction to warmer, more optimal spring temperatures (Wang et al., 2017) possibly accompanied by enough water availability" Why not check this so that your statement can be more definitive?

*Response: We already checked this assumption and confirm later in the results section. We do not want to anticipate later results in the introduction (p.11, l.2).*

• 3.14: "Increased GPP in spring may negatively influence soil moisture and thus GPP during summer (Buermann et al., 2013)." Perhaps elaborate on this a bit more

*Response: We extended the discussion about the article as suggested.*

• 3.20 remove 'equitably'

*Response: Done*

• 5.1: "We consider turbulent fluxes to be biospheric response variables because they are strongly determined by processes in the terrestrial biosphere." I don't quite agree. Latent heat is also a hydrological variable. Is this sentence necessary?

*Response: We agree with the reviewer that latent heat is determined by both, hydrosphere and biosphere. We removed the sentence.*

• 5.3: "The selected variables cover the spatial extent of Europe (latitude 34.5-71.5°N; longitude: -18-60.5°E ) and are regridded on a spatial resolution of 0.25° from 2001 to 2011 in an eight-daily temporal resolution." Perhaps specify that the period is selected as the common period that all datasets cover, a necessary condition for the analysis

*Response: We added a sentence on that: "The temporal extend is selected as it is covered by all datasets."*

• 5.5 I would say 'dominant land cover class'

*Response: We changed "major" in "dominant" as suggested.*

• There are three references to Figure 4 prior to the first reference to Figure 2 (and Figure 3). Usually the figures are ordered according to when they are first referenced in the manuscript text. Please fix as this oversight is irritating.

*Response: We changed the ordering of the Figures accordingly.*

• 5.31 replace 'strategy' with 'methodology'

*Response: Done*

• 8.13: "They are to compute the spatial and temporal distance between res+ and res- ." Awkward sentence, perhaps something is missing. Please rephrase.

*Response: We added the missing word: "They [the centroids] are used to compute ..."*

• Figure 5. Could the authors make the colorbars larger? They are currently illegible.

*Response: Done*

• 10.2: "We find that the GPP response is entirely positive during the shortlasting hydrometeorological spring event (+17:8 Tg C, Tab. 1), while it is mainly negative during the summer (+8:8 Tg C,-49 Tg C, Tab. 5 1). Nonetheless, 18% of the GPP summer losses associated with the RHW in the southern region are instantaneously compensated by over-productive vegetation in the northern latitudes." I'm still not comfortable with how this has been phrased. Please correct me if I have misunderstood but authors argue that the positive GPP anomaly in the high latitudes compensate for the negative GPP anomaly in the mid-latitudes. They are different locations and vegetation types. Yes over the whole EU domain, there is a net positive GPP anomaly but this still doesn't equate to a good outcome over the mid-latitude croplands.

*Response: We apologize for the latter sentence, which can be interpreted as a net positive GPP anomaly of the Russian Heatwave (RHW). Thus, we reformulate the sentence to:*
*"A part of the GPP summer losses (18\%) associated with the RHW in the southern region are instantaneously reduced by over-productive vegetation in the higher latitudes, which are hit by the extreme event."*
*To prevent further missunderstanding, it is important to note that the same hydrometeorological extreme event (the RHW) causes contrasting responses in GPP depending which adjacent ecosystem it hits. We added a sentence for clarification:*
*"Please note, that the carbon balance in summer accounts for the GPP response to the same hydrometeorological extreme event, namely the RHW, which leads to contrasting responses in adjacent regions."*
*We do not average over the whole EU domain, we average only over the region, which was hit by the RHW.*

Perhaps looking in the season after the RHW would provide a sense of the recovery. Perhaps including a map of the net GPP change across both spring and summer would convey that while over the domain as a whole there is a 'GPP compensation' but that there are still regions that could be considered worse off locally.

*Response: We agree with the reviewer, that looking in the season after the RHW provides a sense of the recovery. As noted at p.10, l.8 and p.13, l.3, there are no anomalies after the RHW, which implies a fast recovery of the ecosystems. We already provide maps for spring and summer, separately, their combination does not provide new insights (Fig. 5b,d). The strong differences between adjacent ecosystems can already be seen from the existing maps.*

• 10.10: "integrated annual (spring and summer)" what about winter and autumn?

*Response: We are grateful for the reviewer pointing us to this important point. We removed "spring and summer" as there were neither anomalous events in autumn nor in winter. Thus, the annual integration equals the integration over spring and summer (which we note in the 3rd paragraph of the discussion).*

• I was a bit lost on how to interpret Figure 8 as it shows some soil moisture information that is never discussed. Perhaps the authors could include a statement on how the temperatures correspond to the anomalous soil moisture conditions

*Response: We added a sentence on the soil moisture information (p.12, l.2)*

• 13.1: I'm not sure the simple linear regression model is adding value here? It doesn't tell us anything new.

*Response: Reviewer I (first revision) was very interested in the linear regression model and asked us to better emphasize it. Thus, we moved parts of it from the supplementary to the main text. The model shows that vegetation type along with a latitudinal gradient are the main drivers of gross primary productivity during the heatwave in summer (and not temperature or surface moisture, as expected) (see also p.13, l.6). The objective statistical confirmation of our findings with the model is important and not intuitive to our mind.*

• 14.10: "The absence of events after the summer heatwave which implies a fast recovery of the ecosystems." This would be useful if stated much sooner.

*Response: We added a sentence on the post heatwave recovery by the discussion of Figure 9 (p.12, l.5).*

• 15.25: "Regarding the RHW in particular, compensation effects remain unconsidered in previous studies to the best of our knowledge." Perhaps because the authors consider 'compensation' occurring over two different regions and times as opposed to the same region or at the same time. This is what I find hard to reconcile and I was not satisfied with the revisions made when I commented on this in the first review.

*Response: We removed the sentence of suspect and would like to refer to our response on the concept of compensation before: We argue that summary statistics like aggregation or compensation are valid (1) over the entire region of the heatwave and (2) over time, i.e. the growing season / year to gain meaningful summaries.*

• I'm a bit perplexed that there is both an appendix and supplementary material. Could these perhaps been combined?

*Response: We combined both in the appendix.*

[revised manuscript text omitted]